# Adult patients with autoinflammation of unknown origin partially phenocopy the immune presentation of Still's disease

Rafael Veiga[1,2,12], Leana De Vuyst [3,12], Christophe Poulet [4,5,12], Julika Neumann [3,6], Leoni Bücken [3], Teresa Prezzemolo[3], Mathijs Willemsen [3,6], Steven Vanderschueren[3,7], Patrick Matthys [3], Emna Chabaane[8], Maximilien Fléron[9], Gaël Cobraiville[4], Dominique Baiwir [9], Gabriel Mazzucchelli [9,10], Immunome Project Consortium for Autoinflammatory Disorders (ImmunAID)*, Bruno Fautrel[11], Carine Wouters[3,7,13], Dominique De Seny [4,5,13], Stephanie Humblet-Baron [3,13] ✉ & Adrian Liston [1,2,13] ✉

Autoinflammation of unknown origin remains amongst the most enigmatic of systemic autoinflammatory disorders (SAID), with systemic autoinflammatory symptoms in the absence of a molecular or clinical diagnosis with a recognized SAID. Here, we aim to understand the immunological process behind patients with autoinflammation of unknown origin. We collect samples from 36 patients manifesting recent disease activity across 30 European medical centers, and employ deep immunophenotyping and plasma proteomics to compare to 58 healthy controls and an additional demographically similar cohort comprising 92 SAID patients. Machine-learning approaches identify key immunological changes, including the upregulation of CD38 and HLA across T cell subsets and the upregulation of acute-phase plasma proteins in autoinflammation of unknown origin patients. The immunological traits of these previously poorly characterised patients partially phenocopy Still's disease presentation. Thus, this study identifies potential biomarkers and disease mediators in autoinflammation of unknown origin.

Systemic autoinflammatory disorders (SAID) are a growing class of immune dysregulation disorders, characterized by inappropriate activation of the innate immune system[1]. Distinct from the better studied set of systemic autoimmune disorders owing to the low level of involvement of the adaptive immune system, it is increasingly recognized that SAIDs are a heterogeneous class of disorders, with distinct immunological mechanisms. The recent identification of a genetic basis for many SAIDs has led to a renewed interest in classification and dissection of the underlying pathophysiological causes. In some cases, SAID have a strong genetic background with mutations in

[1]The Babraham Institute, Immunology Programme, Cambridge, UK. [2]University of Cambridge, Department of Pathology, Cambridge, UK. [3]KU Leuven, Department of Microbiology, Immunology and Transplantation, Leuven, Belgium. [4]University of Liège, GIGA institute, Laboratory of Rheumatology, Liège, Belgium. [5]University Hospital of Liège (CHULiège), Department of Rheumatology, Liège, Belgium. [6]VIB Center for Brain and Disease Research, Leuven, Belgium. [7]UZ Leuven, Leuven, Belgium. [8]Université de Paris, Paris, France. [9]GIGA Proteomics Facility, University of Liège, Liège, Belgium. [10]Mass Spectrometry Laboratory, MolSys Research Unit, University of Liège, Liège, Belgium. [11]Sorbonne University, Paris, France. [12]These authors contributed equally: Rafael Veiga, Leana De Vuyst, Christophe Poulet.[13]These authors jointly supervised this work: Carine Wouters, Dominique De Seny, Stephanie Humblet-Baron, Adrian Liston. *A list of members and their affiliations appears in the Supplementary Information. ✉e-mail: stephanie.humbletbaron@kuleuven.be; al989@cam.ac.uk

single genes. However, they can also be of polygenic or multifactorial origin, with environmental influence modulating the phenotype[2]. To date, this process has identified more than 56 Mendelian drivers of SAID, and more than 40 different clinical presentations, with overlapping phenotypic features, genetic associations and immunological drivers associated with the diverse SAIDs[3–5].

The prevalence of SAIDs ranges from 1 in 1000 to 1 in 1,000,000 people, depending on the specific disease, country, and population. Still's disease (encompassing both Adult-onset Still's Disease and systemic Juvenile Idiopathic Arthritis) presents a relatively uniform global distribution with an estimated prevalence of 1–10 cases and 10–100 cases per 1,000,000 population, for the adult and juvenile-onset form, respectively, and occurring at similar rates across diverse populations worldwide[6–8]. Familial Mediterranean Fever exhibits a pronounced ethnic and geographic concentration, with a significantly higher prevalence among specific Mediterranean and Middle Eastern populations. Prevalences vary between 1/250 (Sephardic Jews) and 1/2600 (Arabs). Significant prevalence rates are seen in countries like Greece, Italy, Spain, and Portugal, as well as in diaspora communities where affected populations have migrated[9,10]. Behçet's disease demonstrates a geographic pattern following the ancient Silk Road trading route, with Turkey showing the highest global prevalence rates ranging from 80-420 cases per 100,000 population. The disease is commonly found throughout the Middle East, East Asian countries, and the Mediterranean basin, while becoming progressively rarer in Northern Europe and the Americas with a prevalence between 0.1–7.5 cases per 100,000[11,12]. Beyond these defined conditions, many SAID patients present with undifferentiated autoinflammatory syndromes. A precise epidemiological estimate of these patients, however, presents a significant epidemiological challenge, with prevalence rates likely underreported globally due to inconsistent diagnostic criteria and recognition[13–15].

Several SAIDs have a long-standing recognition as distinct pathologies. In 1930, a first case of what we now recognize as Familial Mediterranean Fever (FMF) was reported[16], a periodic fever syndrome which is one of the most common monogenic SAIDs. By 1997, mutations in the *MEFV* gene, which encodes the protein pyrin and upon activation forms an inflammasome, were identified as the driving force behind FMF[17,18]. Since the 1970s, FMF has typically been treated with colchicine[19,20]. Other SAIDs, however, were long neglected as research subjects, in part due to the rarity of these conditions in individual medical centers. Recent investments in clinical research into SAIDs, with modern research techniques and multi-center consortiums of geneticists, clinicians and scientists in the field, have led to remarkably rapid progress in both understanding the underlying biology and in rational selection of pharmacological interventions. For example, Masters et al. identified a new variant in the *MEFV* gene that leads to Pyrin-Associated Autoinflammation with Neutrophilic Dermatosis (PAAND), a distinct disease from FMF with partially overlapping but unique clinical presentations[21]. The distinctive excessive IL-1β production in PAAND suggested disease control using IL-1β antagonists, demonstrating the value of mechanistic understanding in rational drug selection. Other SAIDs have also seen large advances in understanding the genetic basis, such as Behçet's disease (BD). In Behçet's, with vascular and tissue inflammation, association with the HLA-B*51 gene has been long established, although it accounts for only 20% of the genetic risk[22,23]. More recently, Remmers et al. identified additional associations with the *IL10* and *IL23R-IL12RB2* loci using a genome-wide association study (GWAS) approach[24]. Numerous studies have found associations between various cytokines and BD, influencing both disease onset and severity[25]. Nevertheless, the cellular landscape of BD, and most other SAIDs, remains enigmatic, leaving the full understanding of its pathophysiology incomplete.

Within the spectrum of SAIDs, undifferentiated autoinflammatory diseases (also referred to as 'undefined SAIDs' and as 'syndrome of undifferentiated recurrent fevers' (SURF)) represent a particularly enigmatic group of conditions[26]. These patients have systemic inflammation with increased CRP levels and symptoms of an autoinflammatory disease (fevers, fatigue, rash, arthralgia), in the absence of a molecular diagnosis and without fulfilling the clinical criteria of a distinct polygenic or multifactorial SAID[2,27]. In the current study, we use the term 'autoinflammation of unknown origin' to designate patients who present with recurrent fevers and increased acute phase reactants (CRP > 30 mg/L or SAA > 25 mg/L) occurring over more than 3 months, with no alternative SAID diagnosis during a year-long follow-up period. At present, it is not entirely clear whether autoinflammation of unknown origin represents an individual disease entity or refers to an amalgamation of multiple different conditions. It also may comprise a diverse group of patients with atypical presentations of a defined SAID. This complexity presents significant challenges to patient care, complicating both treatment strategies and accurate prognosis, acknowledging that in broader cohorts of patients with autoinflammation of unknown origin, up to 30% to 50% of the patients remains with no definitive diagnosis[28,29].

Progression on understanding the pathophysiology of inflammation of an unknown origin will rely on multi-center studies and the application of systematic immunological analysis. Here we report on a collaboration across 30 European medical centers, which collected samples from 36 patients with recent disease activity, diagnosed as having autoinflammation of unknown origin. These patients are compared not only to 58 healthy controls, but to an additional 92 SAID patients with recent disease activity and sharing key demographic features, allowing a direct comparison of the immunological state across these conditions. Statistical and machine-learning analysis of high-parameter flow cytometry and mass spectrometry proteomic analysis identify autoinflammation of an unknown origin as a relatively distinct immunological state, with the greatest similarity to Still's disease, partially-phenocopying key immunological changes. These data suggest that autoinflammation of unknown origin is a relatively discrete disease entity at the cellular immune level, and highlight the potential benefit of investigating shared genetic causes and treatment responses to Still's disease patients.

## Results

### Identification of immune phenotypes in autoinflammation of an unknown origin

To develop an insight into the immunological events associated with autoinflammation of an unknown origin, a large number of patients with a spectrum of active systemic autoinflammatory disease across 30 participating European hospitals were recruited. Patients were enrolled with recent disease activity and diagnosed according to the respective SAID diagnostic criteria (Table 1, with expanded individual clinical data available as Supplementary Spreadsheet 1), with peripheral blood mononuclear cells (PBMC) collected. Our study included 32 patients classified as having autoinflammation of unknown origin, and an additional 92 patients with well-defined SAIDs following a molecular diagnosis or internationally approved clinical criteria. The selection of SAIDs was based on the availability of sufficient samples for statistical comparison and on a similar demographic profile compared to the group autoinflammation of unknown origin. This resulted in 34 patients with Still's disease (a polygenic/multifactorial SAID characterized by persistent high fevers, rash and joint symptoms), 35 patients with FMF (a prototypic monogenic hereditary fever syndrome) and 23 patients with Behçet's disease (a polygenic/multifactorial SAID characterized predominantly by aphthosis, ocular symptoms and vasculitis). Patients diagnosed in rare disease categories or disease categories with significantly different demographic profiles were excluded from the analysis due to insufficient power to run statistical analysis. An additional 58 healthy individuals, recruited simultaneously, were included, for a total of 186 individuals. Each set of

**Table 1 | Summary statistics for disease differences**

| Flow cytometry: average area under the receiver operating characteristic curve | | | | | |
|---|---|---|---|---|---|
| | Healthy | AUO | Behcet | FMF | Still's |
| Healthy | NA | 0.92 (0.86–0.98) | 0.77 (0.64–0.89) | 0.78 (0.68–0.89) | 0.94 (0.87–1.01) |
| AUO | 0.92 (0.86–0.98) | NA | 0.79 (0.64–0.93) | 0.82 (0.67–0.96) | 0.79 (0.67–0.91) |
| Behcet | 0.77 (0.64–0.89) | 0.79 (0.64–0.93) | NA | 0.78 (0.64–0.93) | 0.93 (0.85–1.01) |
| FMF | 0.78 (0.68–0.89) | 0.82 (0.67–0.96) | 0.78 (0.64–0.93) | NA | 0.93 (0.86–0.99) |
| Still's | 0.94 (0.87–1.01) | 0.79 (0.67–0.91) | 0.93 (0.85–1.01) | 0.93 (0.86–0.99) | NA |
| Flow cytometry: Euclidean distance between population averages | | | | | |
| | Healthy | AUO | Behcet | FMF | Still's |
| Healthy | NA | 6.17 | 4.86 | 4.45 | 8.90 |
| AUO | 6.17 | NA | 6.59 | 6.12 | 5.14 |
| Behcet | 4.86 | 6.59 | NA | 4.26 | 8.87 |
| FMF | 4.45 | 6.12 | 4.26 | NA | 8.10 |
| Still's | 8.90 | 5.14 | 8.87 | 8.10 | NA |
| Proteomics: average area under the receiver operating characteristic curve | | | | | |
| | Healthy | AUO | Behcet | FMF | Still's |
| Healthy | NA | 0.94 (0.89–1.00) | 0.85 (0.75–0.96) | 0.93 (0.86–1.01) | 0.97 (0.94–1.00) |
| AUO | 0.94 (0.89–1.00) | NA | 0.64 (0.45–0.83) | 0.72 (0.59–0.86) | 0.65 (0.46–0.84) |
| Behcet | 0.85 (0.75–0.96) | 0.64 (0.45–0.83) | NA | 0.71 (0.54–0.88) | 0.95 (0.89–1.00) |
| FMF | 0.93 (0.86–1.01) | 0.72 (0.59–0.86) | 0.71 (0.54–0.88) | NA | 0.90 (0.81–0.98) |
| Still's | 0.97 (0.94–1.00) | 0.65 (0.46–0.84) | 0.95 (0.89–1.00) | 0.90 (0.81–0.98) | NA |
| Proteomics: Euclidean distance between population averages | | | | | |
| | Healthy | AUO | Behcet | FMF | Still's |
| Healthy | NA | 8.31 | 8.11 | 7.24 | 10.41 |
| AUO | 8.31 | NA | 7.24 | 5.98 | 5.83 |
| Behcet | 8.11 | 7.24 | NA | 5.93 | 9.95 |
| FMF | 7.24 | 5.98 | 5.93 | NA | 8.46 |
| Still's | 10.41 | 5.83 | 9.95 | 8.46 | NA |

PBMCs was analyzed by high-parameter flow cytometry and gated for 208 defined immunological subsets, covering the major leukocyte populations present in PBMCs. This dataset, available as a resource, was used to assess the immune phenotype of patients presenting with autoinflammation of an unknown origin and other well-defined systemic autoinflammatory diseases.

Before performing a multi-disease analysis, we first compared the immune status of patients with autoinflammation of unknown origin to healthy individuals. Using a multivariate logistic regression, adjusted for sex and age, we identified the immune parameters significantly associated with autoinflammation of unknown origin (Fig. 1A and Supplementary Fig. 1). The first 40 parameters added the greatest explanatory power to discriminate between autoinflammation of unknown origin and healthy individuals (Fig. 1B). Of these, the strongest effects were increases in CD38+ subsets (including within the CD27- CD8 + T cell subset, within the total CD8 + T cell population, within the NKT cell population, within CD16 + NK cells, within DN T cells, and within NK cells), decreased IgM + IgD+ memory B cells, decreased BAFF-R$^{high}$ naïve B cells (among IgM+IgD+ naïve B cells and IgM$^{low}$ IgD + naïve B cells) and decreased HLA-DR + cells within the CD16-CD56$^{dim}$ NK population (Fig. 1C). The aggregate of data in the 40 highly associated parameters was highly discriminatory against healthy individuals, with an AUC of 0.83 (Fig. 1D). While the full set of immune parameters poorly separated autoinflammation of unknown origin patients from healthy individuals (Fig. 1E), indicative of non-disease variation being the primary driver of diversity in the dataset, the top parameters (from Fig. 1A) discretely separated the populations (Fig. 1F). Notably, while these immunological changes were highly discriminatory between the autoinflammation of unknown origin

patients and the healthy individuals, the strongest parameters had poor discrimination with other systemic inflammation patient sets in our data set, with the Still's disease population in particular showing similar or greater changes in each parameter (Fig. 1C). Together, these data suggest that autoinflammation of unknown origin is a discrete immunological condition, or a collection of different conditions with a similar immune signature with patients clustered together on the basis of consistent changes compared to healthy individuals.

To determine if the immunological signature observed was driven by long-lasting effects of treatment, we subdivided the patients with autoinflammation of unknown origin into individuals who were treated and those who were untreated. For each of the key associated immunological parameters identified (Fig. 1C), similar effect sizes were observed between the untreated and treated patients (Fig. 2A). At a global level (Fig. 2B) and specifically within the associated parameters (Fig. 2C), the treated and untreated patients were more similar to each other than they were to healthy controls, indicating that the signature was driven by disease status, rather than treatment.

**Immunological changes across systemic autoinflammatory diseases**

The identification of an immunological profile shared across autoinflammation of unknown origin patients suggests a common underlying immunological process. We next sought to determine whether this immunological profile was shared with other systemic inflammatory diseases or constituted a unique immunological entity. Based on the referred to systemic autoinflammatory diseases, with similar demographic profiles, in the participating hospitals, we had patient datasets available with Still's disease, FMF and Behçet's disease.

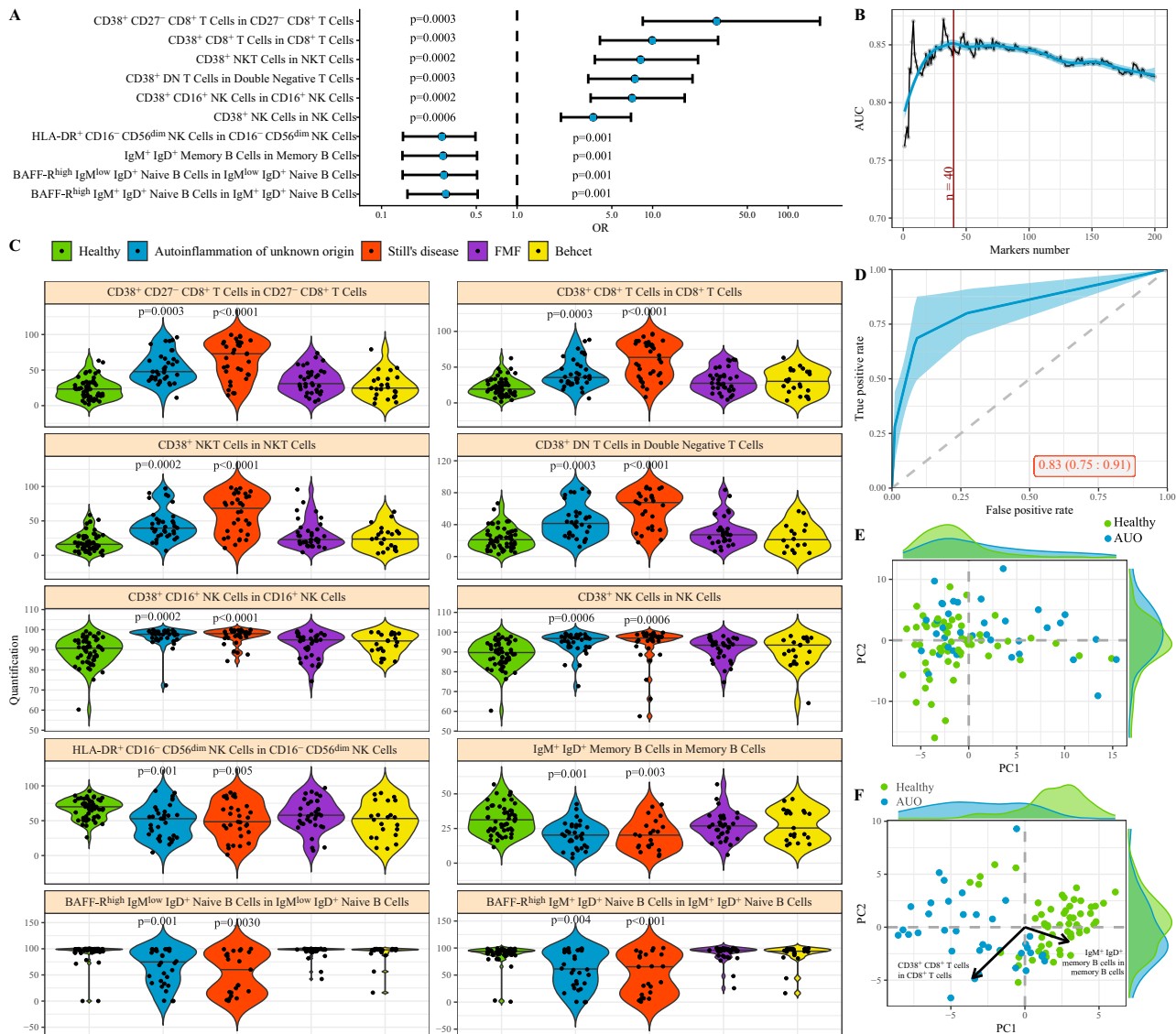

**Fig. 1 | Immunological parameters associated with autoinflammation of unknown origin.** Patients with autoinflammation of unknown origin ($n = 36$) were compared against healthy individuals ($n = 58$) by multivariate logistic regression of immunological parameters. Data from patients with Still's disease ($n = 34$), FMF ($n = 35$) and Behçet ($n = 23$) are shown as reference data only. **A** Odds ratio and 95% confidence interval (point and error bars) of highly-associated cell population frequency changes in patients with autoinflammation of unknown origin in relation to healthy individuals. Estimated by multivariable logistic regression adjusted by sex and age. *P*-values multiple test adjusted using Benjamini–Hochberg. **B** Average and 95% confidence (blue line and blue band) interval of 200 times 10 fold cross-validation to evaluate a sufficient number of best cell populations, based on the ability to adequately discern between autoinflammation of unknown origin and healthy individuals. **C** Frequency for highly-associated cell populations for auto-inflammation of unknown origin in relation to healthy individuals. Each dot represents a patient, and each color represents a condition. Bar indicates median, the

violin plot indicates data density. Adjusted p values for each disease compared to healthy are indicated where $p < 0.05$ (Mann–Whitney u test, two-tailed). **D** Average ROC curve with 95% confidence interval (blue line and blue band) of 10-fold cross-validation for autoinflammation of unknown origin in relation to healthy individuals. ROC calculated using multi-variable logistic regression, adjusted by sex and age, considering the 40 cell populations with the highest explanatory contribution. Area under the ROC curve and confidence interval indicated on the graph. **E** First two PCA components of all cell populations in the dataset. Each dot represents an individual, and each color represents a condition. Histograms show the distribution of values in autoinflammation of unknown origin and healthy individuals. **F** First two PCA components of 40 cell populations most highly associated for divergence between autoinflammation of unknown origin and healthy individuals. Histograms show the distribution of values in autoinflammation of unknown origin and healthy individuals. The two arrows show the direction of distinct highly associated cell populations. AUO, autoinflammation of unknown origin.

We first performed a disease vs healthy multivariate logistic regression for each disease in isolation. Still's disease presented with a strong immunological phenotype (Supplementary Fig. 2), with the greatest change observed in parameters associated with elevated CD38 expression (in CD8 + T cells, DN T cells, NKT cells, CD16 + NK cells and CD27- CD4 + T cells) and a higher fraction of HLA-DR+ cells (within CD4 + T cells, and CD4 + T cell subsets). FMF and Behçet's, by contrast, presented with weaker immunological differences from the

healthy population (Supplementary Figs. 3, 4), with similar immunological features identified. Together, these separate analyses broadly suggested divergent programs in the systemic inflammatory diseases analyzed, with autoinflammation of unknown origin associated with similar parameters to Still's disease, while FMF and Behçet's had distinct, and weaker, immunological changes.

To formally compare between diseases, we turned to a multi-disease machine learning approach. To identify the discriminators

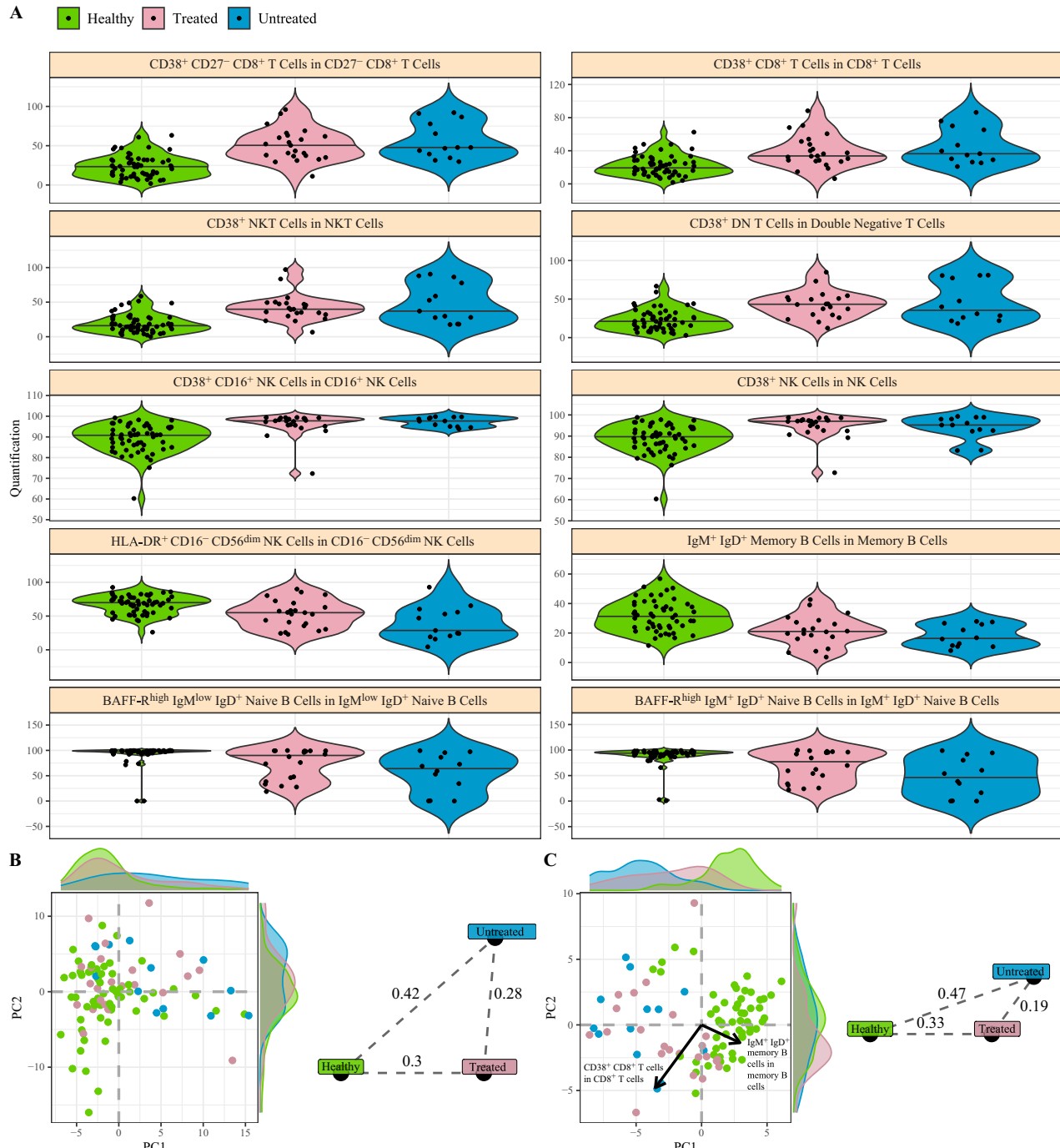

**Fig. 2 | Immunological fingerprint of autoinflammation of unknown origin independent of treatment status.** Patients with autoinflammation of unknown origin were stratified into untreated (*n* = 13) and treated (*n* = 23). Patients were compared to healthy individuals (*n* = 58) for associated immunological parameters. **A** Frequency for highly-associated cell populations for autoinflammation of unknown origin in relation to healthy individuals, with patients stratified based on treatment status. No differences between treated and untreated patients reached *p* < 0.05 (Mann-Whitney u test, two-tailed). Bar indicates median, violin plot indicates data density. **B** First two PCA components of all cell populations in the

dataset. Each dot represents an individual, and each color represents a disease and treatment status. Histograms show the distribution of values in autoinflammation of unknown origin and healthy individuals. Euclidean distances displayed for the population averages. **C** First two PCA components of 40 cell populations most highly associated for divergence between autoinflammation of unknown origin and healthy individuals. Histograms show the distribution of values in autoinflammation of unknown origin and healthy individuals. The two arrows show the direction of distinct highly associated cell populations. Euclidean distances displayed for the population averages.

between diseases, we built a Random Forest algorithm without healthy individuals in the training set, while including healthy individuals in the analysis post-parameter selection. The model identified a complex set of immunological parameters which, when used together, provided

discriminatory capacity between diseases (Supplementary Fig. 5). The developed model was tested for discriminatory capacity via Receiver Operating Characteristic (ROC) curves in pair-wise comparisons (Fig. 3). The model varied in its discrimination power depending on the

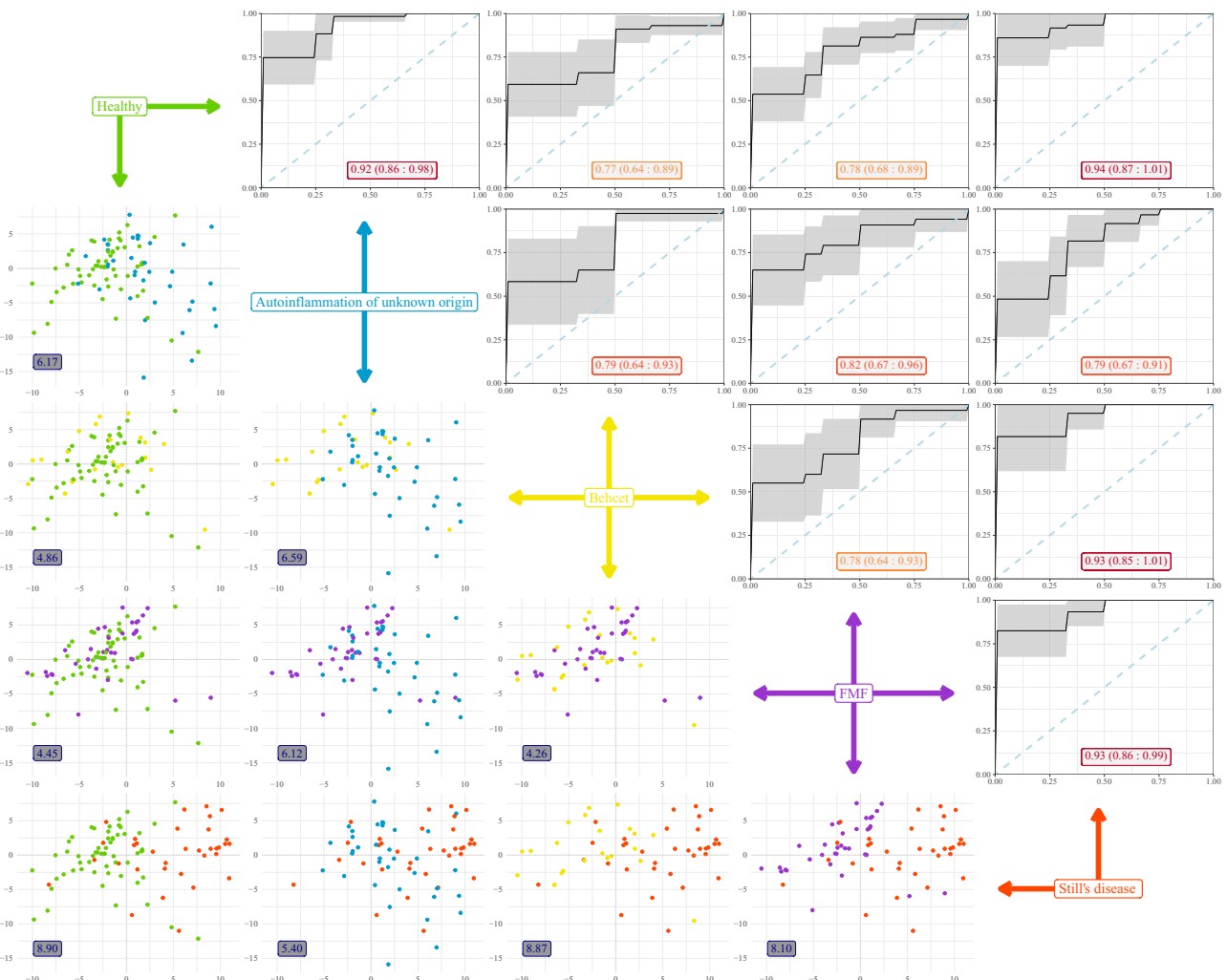

**Fig. 3 | Machine learning-led multi-disease comparison clusters autoin-flammation of unknown origin patients with Still's disease patients.** A multi-disease comparison was performed using immune phenotypes from patients with autoinflammation of unknown origin ($n = 36$), Still's disease ($n = 34$), FMF ($n = 35$) and Behçet ($n = 23$). Healthy individuals are shown as reference data only, not being used to build immuno-discriminating models. In the upper-right panels, a Random Forest algorithm was built to discriminate between diseases, with the Receiver Operating Characteristic (ROC) curve shown for each pairwise comparison of disease-disease and healthy-disease. The black line represents the average ROC curve obtained from 10-fold cross-validation. The shaded gray area indicates the 95% confidence interval. The numeric labels within the plots show the average area under the ROC curve (AUC) and the 95% confidence interval for each comparison. In the lower-left panels, individuals are plotted on the first two principal components for the respective comparison. The numeric labels within the plots indicate Euclidean distances displayed for the population averages of the pairwise comparison.

disease comparisons made. For autoinflammation of unknown origin, discrimination power was highest against healthy individuals (0.92), with poorer discrimination against the other inflammatory conditions (0.79-0.82). Still's disease had strong discrimination power between all conditions, except autoinflammation of unknown origin, while FMF and Behçet's showed relatively weak discrimination, except in comparison to Still's disease (Fig. 3 and Table 1). Using an independent approach, with a PCA-based comparison, again built only on the four disease groups, individuals formed two main clusters – one including healthy, FMF and Behçet's individuals (reflect by low Euclidean distances between these three conditions), and one including Still's disease and autoinflammation of unknown origin individuals (with low Euclidean distance), with comparisons between any two pairs cross-cluster showing elevated immunological distance (Fig. 3 and Table 1). Together, these data demonstrate that immune phenotype is effective in discriminating autoinflammation of unknown origin and Still's disease from healthy individuals and patients with FMF and Behçet's, while disease-specific signatures were weaker components of the variation observed.

The immunological parallels between autoinflammation of unknown origin and Still's disease was also observed when looking at the key explanatory immune parameters in isolation (Fig. 4). For multiple parameters, changes between Still's disease and autoin-flammation of unknown origin moved in tandem, while individuals in the other diseases remained unchanged. BAFF-R expression (in the total B cell population and multiple subsets of naïve and transitional B cells) behaved in this manner. For other parameters, chiefly CD38 expression (in CD27- CD8 + T cells and DN T cells), the phenotype was strongest in Still's disease, with autoinflammation of unknown origin moving in a similar direction but to a lower extent. The effect of HLA-DR expression (across multiple T cell subsets) was largely seen in Still's disease. Together, these individual immunological changes, among the most inter-disease discriminatory in nature, support the conclusion that autoinflammation of unknown origin partially phenocopies the cellular immunological presentation of Still's disease, while being a distinct immunological entity to the other systemic inflammatory conditions tested.

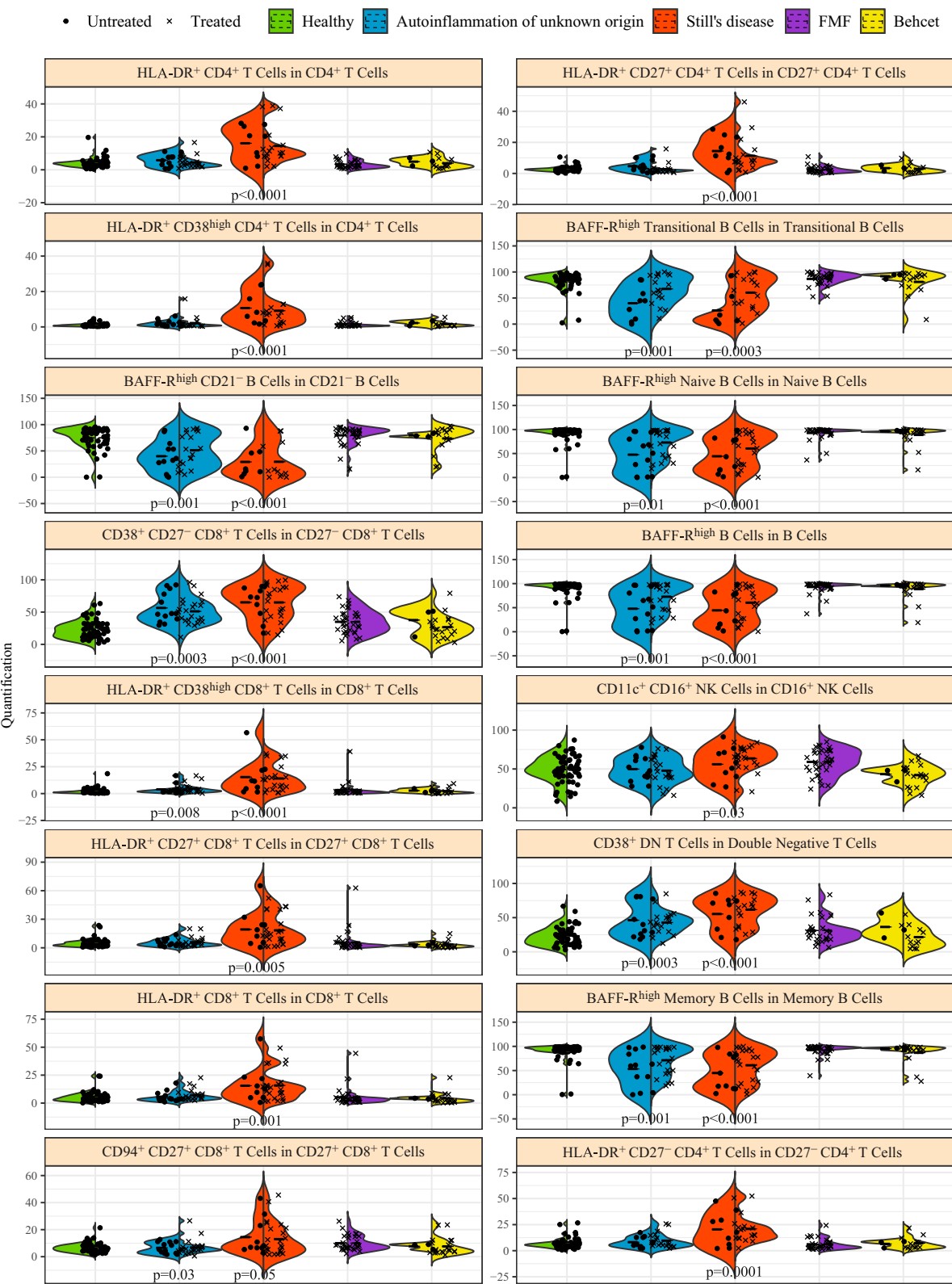

**Fig. 4 | Key immunological features driving machine learning-led disease identification.** A multi-disease comparison was performed using a Random Forest algorithm to identify immune characteristics with discriminating potential between patients with autoinflammation of unknown origin ($n = 36$), Still's disease ($n = 34$), FMF ($n = 35$) and Behçet ($n = 23$). Healthy individuals were not used in the model generation, and are shown as reference data only. Frequency for the 16 highest associated cell populations for disease discrimination. Each dot represents a patient, and each color represents a condition. Bar indicates median, violin plot indicates data density. For each condition, treated and untreated patients are separately indicated. $p$-values for each disease compared to healthy are indicated where $p < 0.05$ (Mann-Whitney u test, two-tailed). No differences between treated and untreated patients reached $p < 0.05$.

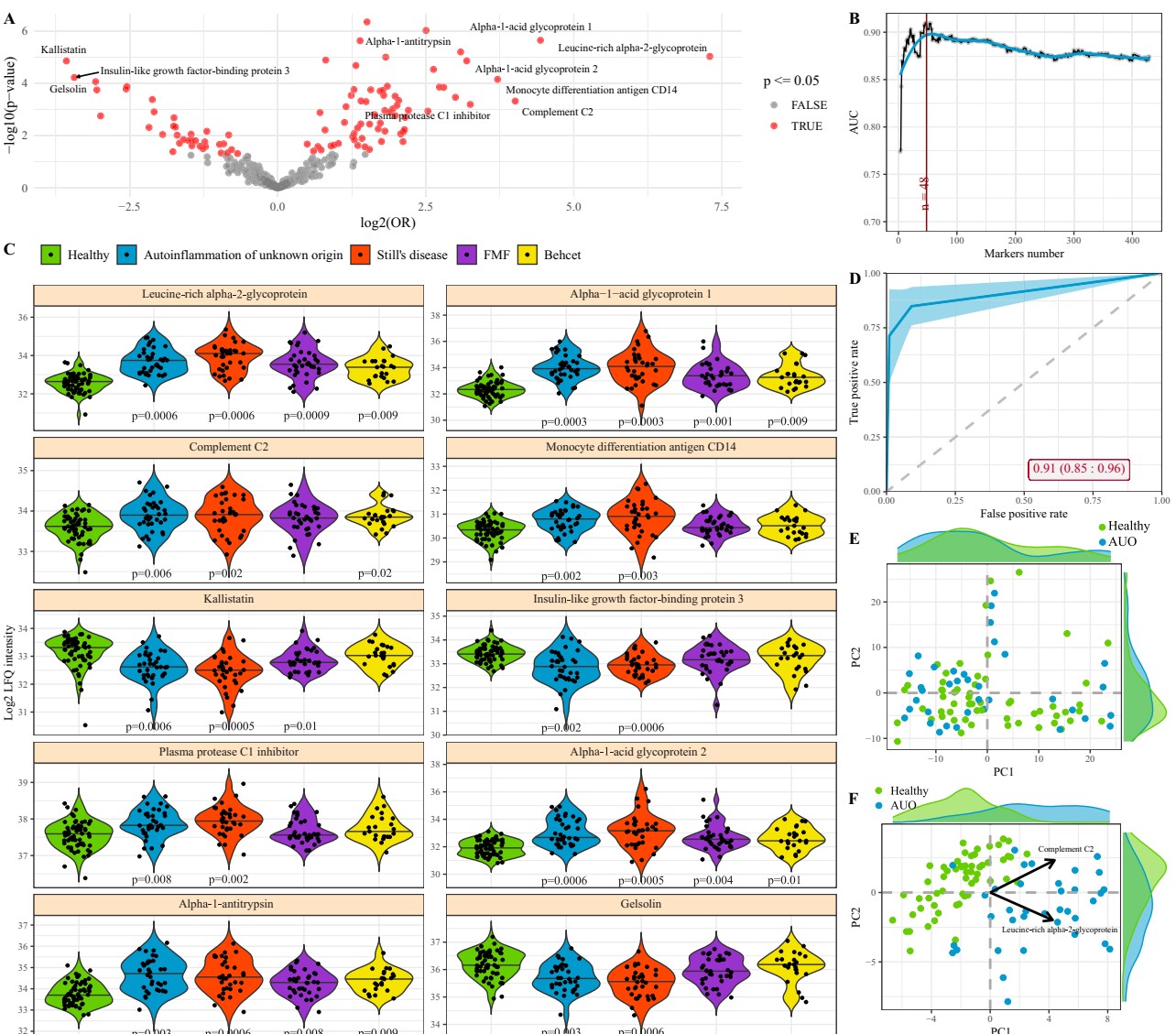

**Fig. 5 | Plasma proteomic changes associated with autoinflammation of unknown origin.** Patients with autoinflammation of unknown origin (*n* = 36) were compared against healthy individuals (*n* = 58) by multivariate logistic regression of plasma proteomics. Data from patients with Still's disease (*n* = 34), FMF (*n* = 35) and Behçet (*n* = 23) are shown as reference data only. **A** Volcano plot of Odds ratio and *p*-value of highly-associated protein changes in patients with autoinflammation of unknown origin in relation to healthy individuals. Estimated by multivariable logistic regression adjusted by sex and age. **B** Average and 95% confidence interval (blue line and blue band) of 200 times 10 fold cross-validation to evaluate a sufficient number of proteomic markers, based on the ability to adequately discern between autoinflammation of unknown origin and healthy individuals. **C** Detection of highly-associated proteins for autoinflammation of unknown origin in relation to healthy individuals. Each dot represents a patient, and each color represents a condition. Adjusted *p*-values for each disease compared to healthy are indicated

where *p* < 0.05 (Mann-Whitney u test, two-tailed). Bar indicates median, violin plot indicates data density. **D** Average ROC curve with 95% confidence interval (blue line and blue band) of 10-fold cross-validation for autoinflammation of unknown origin in relation to healthy individuals. ROC calculated using multivariable logistic regression, adjusted by sex and age, considering the 48 plasma proteins with the highest explanatory contribution. Area under the ROC curve and confidence interval indicated on graph. **E** First two PCA components of all plasma proteins in the dataset. Each dot represents an individual, and each color represents a condition. Histograms show the distribution of values in autoinflammation of unknown origin and healthy individuals. **F** First two PCA components of 48 plasma proteins most highly associated for divergence between autoinflammation of unknown origin and healthy individuals. Histograms show the distribution of values in autoinflammation of unknown origin and healthy individuals. The two arrows show the direction of distinct highly associated plasma proteins.

## An inflammatory proteomic signature in autoinflammation of an unknown origin

To test the potential immunological parallels between autoinflammation of an unknown origin and Still's disease, we turned to an independent biological measure, using mass spectrometry of plasma samples from our patient cohort. First comparing patients with autoinflammation of unknown origin to healthy individuals using a multivariate logistic regression, we found diverse plasma protein changes

significantly associated with autoinflammation of unknown origin (Fig. 5A and Supplementary Fig. 6), with the first 48 parameters driving explanatory discrimination (Fig. 5B). The strongest effects were the upregulation of acute phase inflammatory proteins, including Leucine−rich alpha−2−glycoprotein, Alpha−1−acid glycoprotein 1 and 2, and Complement C2, together with anti-inflammatory responders Alpha−1−antitrypsin and plasma protease C1 inhibitor (Fig. 5C). Fewer proteins were downregulated, with Kallistatin, Gelsolin and IGFBP-3

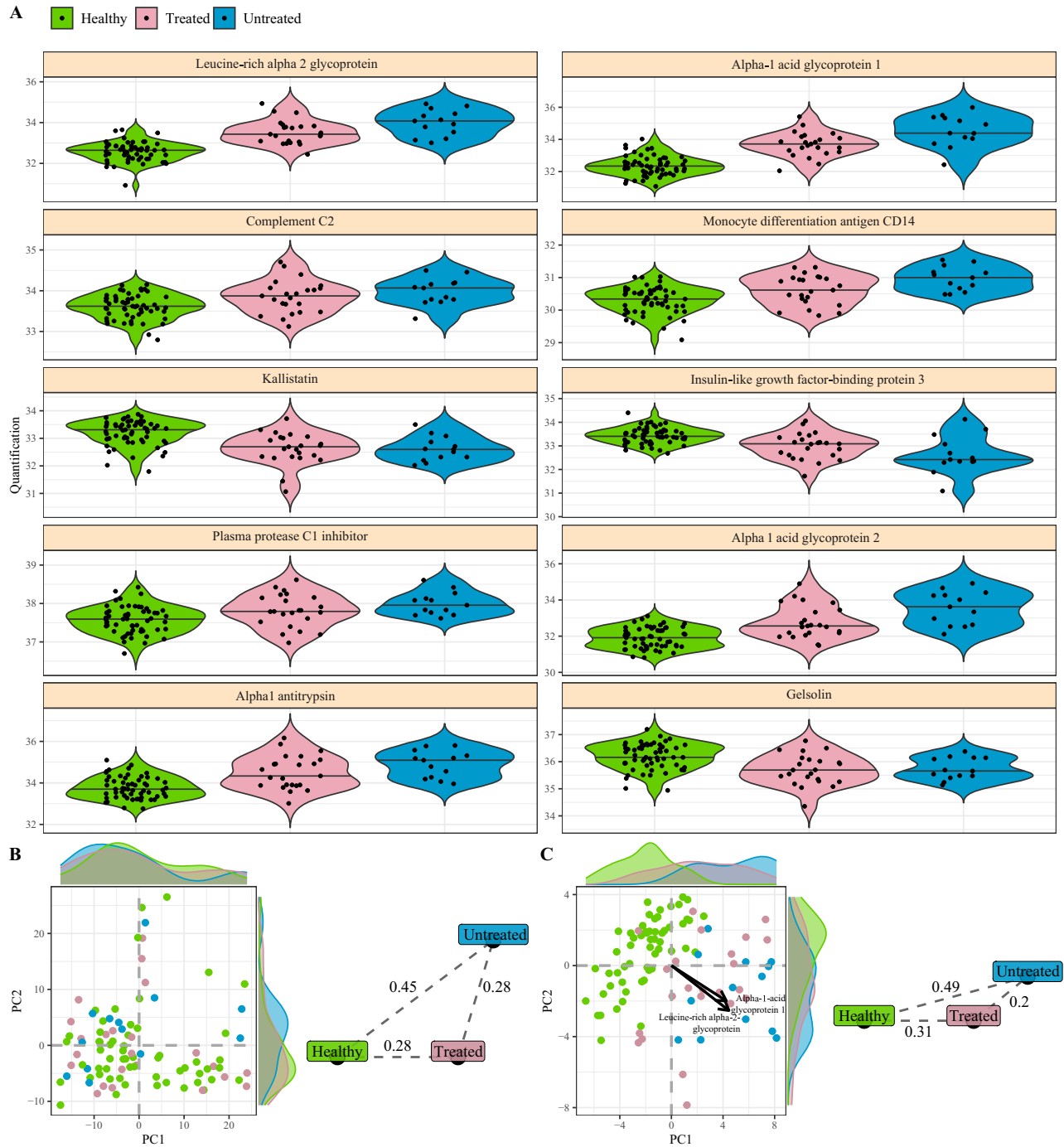

**Fig. 6 | Plasma proteomic changes associated with autoinflammation of unknown origin are independent of treatment status.** Patients with autoinflammation of unknown origin were stratified into untreated ($n = 13$) and treated ($n = 23$). Patients were compared to healthy individuals ($n = 58$) for associated plasma proteins. **A** The LFQ intensity of highly-associated plasma protein changes for autoinflammation of unknown origin in relation to healthy individuals, with patients stratified based on disease status. No differences between treated and untreated patients reached $p < 0.05$ (Mann-Whitney u test, two-tailed). Bar indicates median, violin plot indicates data density. **B** First two PCA components of all plasma proteins populations in the dataset. Each dot represents an individual, and each color represents a disease and treatment status. Histograms show the distribution of values in autoinflammation of unknown origin and healthy individuals. Euclidean distances displayed for the population averages. **C** First two PCA components of 48 plasma proteins most highly associated for divergence between autoinflammation of unknown origin and healthy individuals. Histograms show the distribution of values in Still's disease and healthy individuals. The two arrows show the direction of distinct highly associated plasma proteins. Euclidean distances displayed for the population averages.

notable as proteins with protective anti-inflammatory properties that were found at lower levels in patients. Combined, these 48 proteins were highly discriminatory, with an AUC of 0.91 (Fig. 5D), and (unlike the total plasma proteome, Fig. 5E) provided discrete separation of

patients from healthy control (Fig. 5F). These changes were not driven by treatment regimes, with highly similar profiles of treated and untreated patients within the associated parameters (Fig. 6). As observed in the flow cytometry data, many of these proteomic changes

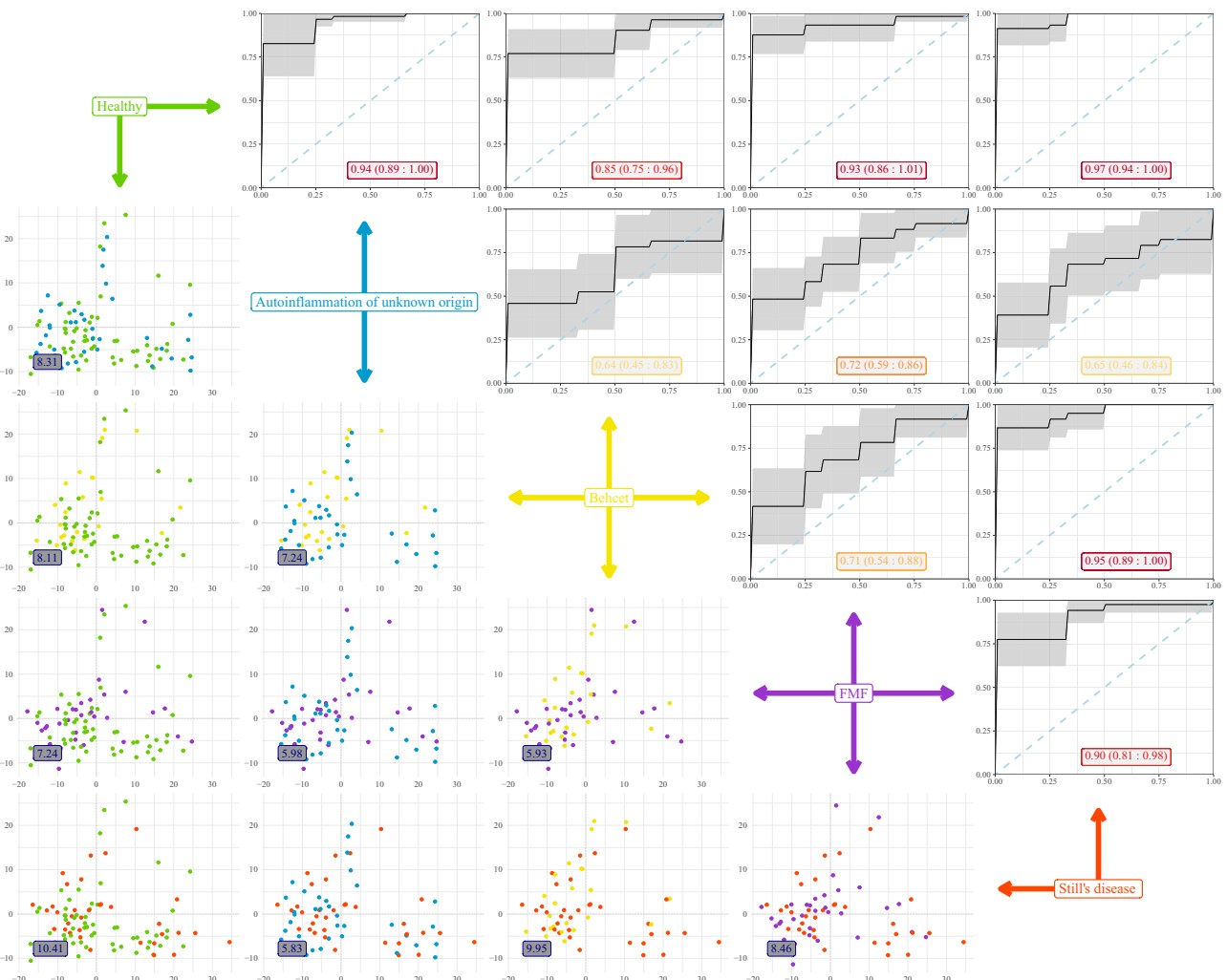

**Fig. 7 | Machine learning-led multi-disease comparison distinguishes Still's disease with autoinflammation of unknown origin intermediate to other inflammatory disease profiles.** A multi-disease comparison was performed using plasma proteomics data from patients with autoinflammation of unknown origin (*n* = 36), Still's disease (*n* = 34), FMF (*n* = 35) and Behçet (*n* = 23). Healthy individuals are shown as reference data only, not being used to build immuno-discriminating models. In the upper-right panels, a Random Forest algorithm was built to discriminate between diseases, with the Receiver Operating Characteristic (ROC) curve shown for each pairwise comparison of disease-disease and healthy-disease. The black line represents the average ROC curve obtained from 10-fold cross-validation. The shaded gray area indicates the 95% confidence interval. The numeric labels within the plots show the average area under the ROC curve (AUC) and the 95% confidence interval for each comparison. In the lower-left panels, individuals are plotted on the first two principal components for the respective comparison. The numeric labels within the plots indicate Euclidean distances displayed for the population averages of the pairwise comparison.

in patients of autoinflammation of an unknown origin were also observed in other autoinflammatory patients, in particular those with Still's disease (Fig. 5C).

To determine the uniqueness of the proteomic signature of autoinflammation of an unknown origin, we replicated the analysis for each of the other autoinflammatory diseases. Highly similar proteomic profiles were observed as discriminating healthy individuals from patients with Still's disease (Supplementary Fig. 7), FMF (Supplementary Fig. 8) or Behçet's (Supplementary Fig. 9). Using a multi-disease machine learning approach, a proteomics signature was identified with the highest discriminatory capacity between diseases (Supplementary Fig. 10). This set of discriminatory proteins was most effective at distinguishing patients from healthy individuals (ROC 0.85–0.97, with autoinflammation of an unknown origin at 0.94) (Fig. 7 and Table 1). It also effectively discriminated between Still's disease and both Behcet and FMF (0.95, 0.90), with weaker explanatory capacity between other pairwise comparisons. Autoinflammation of unknown origin, by this measure, was only poorly distinguished from the other

autoinflammatory diseases, with the closest profile (as measured by Euclidean distance) being that of Still's disease (Fig. 7 and Table 1).

Analysis of the key explanatory proteomics parameters across diseases reinforces the designation of Still's disease as a key driver of disease differentiation (Fig. 8). Most of the protein changes observed were either pan-disease (such as CRP or S100A9), with inflammatory markers elevated in each disease, often with Still's disease showing the highest degree of change, or largely restricted to Still's disease, with patients showing upregulation of inflammatory proteins such as Leukotriene A-4 hydrolase or LILRA3 and downregulation of extracellular matrix components such as lumican (Fig. 8). Of the upregulated proteins that broke this pattern, IgGFc-binding protein and plasma protease C1 inhibitor, both were shared across autoinflammation of unknown origin and Still's disease only, consistent with the prior conclusions that autoinflammation of unknown origin parallels Still's disease in several key biomarkers. This trend was observed using diagnostic quantification of the classical Still's disease plasma markers of ferritin, IL-18 and CXCL9, where patients with autoinflammation of

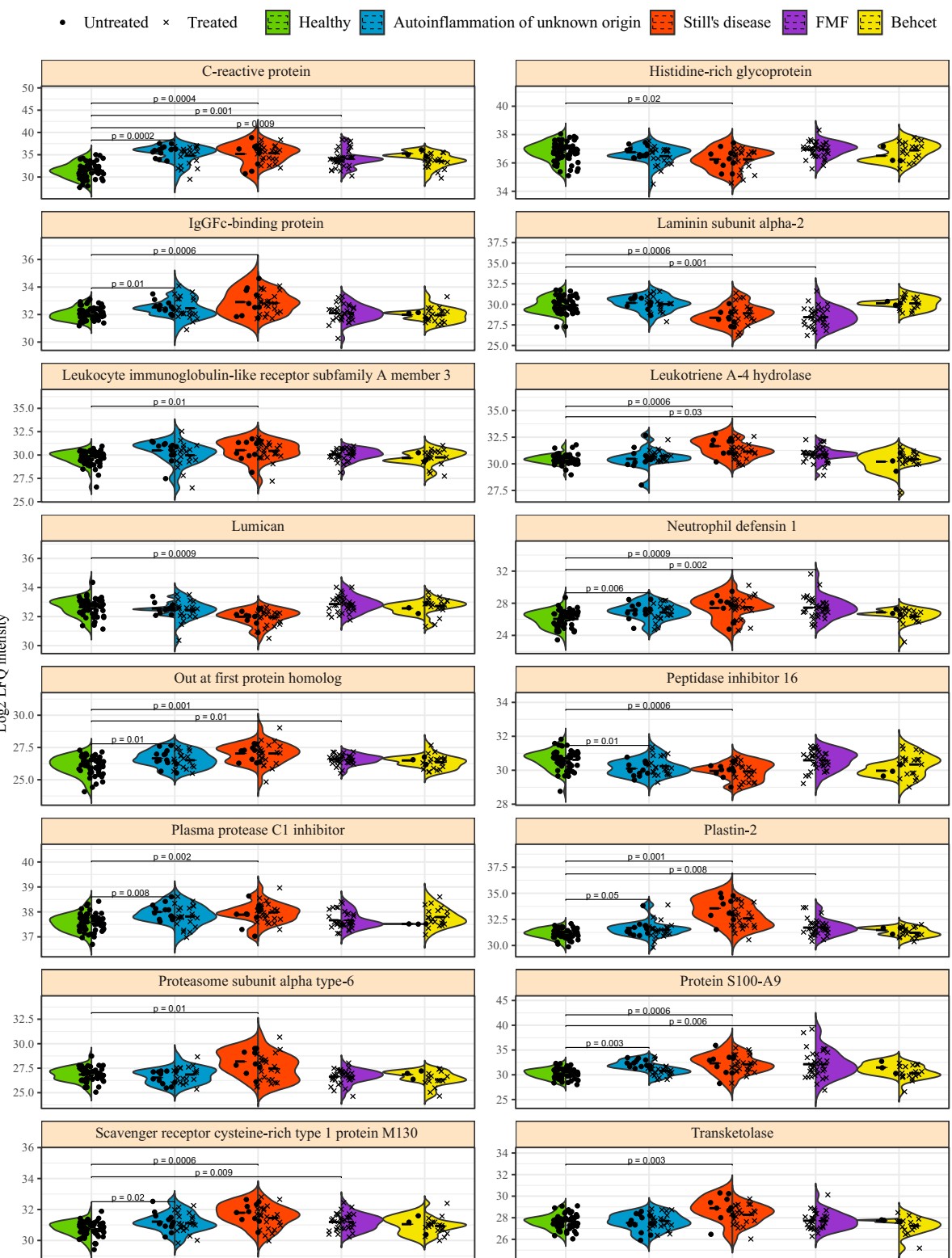

**Fig. 8 | Key immunological features driving machine learning-led disease identification.** A multi-disease comparison was performed using a Random Forest algorithm to identify immune characteristics with discriminating potential between patients with autoinflammation of unknown origin ($n = 36$), Still's disease ($n = 34$), FMF ($n = 35$) and Behçet ($n = 23$). Healthy individuals were not used in the model generation, and are shown as reference data only. Intensity for the 16 highest associated plasma proteins for disease discrimination. Each dot represents a patient, and each color represents a condition. Bar indicates median, violin plot indicates data density. For each condition, treated and untreated patients are separately indicated. Adjusted $p$-values for each disease compared to healthy are indicated where $p < 0.05$ (Mann-Whitney u test, two tailed). Protein S100A9 demonstrated significant differences between treated and untreated in autoinflammation of unknown origin patients, while all other treated/untreated comparisons were not significant at the $P < 0.05$ threshold.

unknown origin showed an intermediate phenotype between that of healthy and Still's disease patients (Supplementary Fig. 11).

## Discussion

The number and scope of SAIDs continue to grow thanks to increased clinical recognition and collaborations worldwide, as well as continuous advances in molecular diagnosis. As SAIDs are a heterogeneous family of rare disorders with significant overlap in clinical characteristics, nonspecific symptoms, and multisystem involvement, a rapid and precise diagnosis is often challenging. For instance, recent publications have highlighted the continuum between Still's disease and other systemic inflammatory disorders and distinguishing Still's disease from other SAIDs can be challenging, considering monogenic autoinflammatory conditions, Schnitzler syndrome or Sweet syndrome in addition to sarcoidosis or spondylarthritis[30,31]. Similarly, Behçet's disease shares features with inflammatory bowel disease and other types of vasculitis, complicating its diagnosis[32]. As a result, long-term follow-up is often required to identify diagnostic distinctness, especially in systemic conditions.

Understanding the genetic and immunological drivers of these diseases could aid in diagnosis. Surprisingly, unlike autoimmune diseases, which are generally multigenic, many SAIDS follow Mendelian inheritance patterns, providing the potential for gene-informed clinical diagnosis. Despite the attractiveness of genetics as a simple diagnostic, in practice, even the Mendelian disorders show complex genotype-phenotype relationships, with single clinical entities having multiple different genetic causes and mutations in the same genes being the drivers in multiple different disorders[33]. This genetic pleiotropy and heterogeneity complicate accurate diagnosis, and there is thus a need for other biomarkers to aid diagnosis. As a functional driver of clinical presentation, immunological changes are among the most attractive potential biomarkers for SAIDs. This is especially in cases of autoinflammation of unknown origin, where the diagnosis is currently based on the slow exclusion of other SAIDs, along with the use of classification criteria and FDG-PET scans. While the involvement of certain immune players is known for some SAIDs[34], the detailed immunophenotyping of other patient sets remains incomplete, and the degree to which immunological changes will be able to aid diagnosis remains undetermined.

This study aimed to facilitate the diagnostic process of one of the most enigmatic SAIDs, undifferentiated autoinflammatory diseases (autoinflammation of unknown origin in this manuscript), by defining the specific immune phenotype. To thoroughly study the immune signature of autoinflammation of unknown origin, large cohorts of patients, including untreated patients, are necessary, since defining immunological drivers is further complicated by the treatments patients receive. In addition to testing patients naïve of treatment, it is important to include relevant controls, rather than only healthy individuals. Pan-inflammation markers such as CRP are diagnostically useful in identifying patients with inflammation, but have poor discriminatory power to provide a finer level diagnosis. The comparative groups used here were SAID patients with similar demographic characteristics, allowing a cross-condition comparison.

By comparing patients with different SAIDs (including autoinflammation of unknown origin, Still's disease, Behçet's disease and FMF) to each other and to healthy individuals, valuable information could be identified, distinguishing disease states. Multiple immunological signatures were identified across the diseases, with the key discovery being the partial immunophenocopying observed between autoinflammation of unknown origin and Still's disease. Previous studies have demonstrated altered B cell subsets in Still's disease patients, compared to healthy individuals[34,35]. Our data, however, additionally suggest an important role for T cells, as supported by Myachikova et al.[36] T cells in the synovial fluid have already been shown to play an important role in the disease phenotype of juvenile idiopathic arthritis[37], and CD4 + and CD8 + T cells locally play an important role in tissue-specific inflammation and disease relapse[38]. Our findings specifically showed that CD38 and HLA-DR expression in multiple T cell subsets was associated with Still's disease. Jung et al. have also reported elevated HLA-DR expression in Still's disease patients, though they also noted decreased CD4 + and CD8 + cells[39]. Interestingly, a similar phenotype to our observation, expansion of CD38 + HLA-DR+ cells within CD4 + T cells, CD8 + T cells and NK cell populations, has been associated with macrophage activation syndrome, a life-threatening complication of Still's disease[40]. This elevated CD38 and HLA-DR expression in patients with autoinflammation of unknown origin, although to a lower extent than that observed in Still's disease patients, underscores the need to further explore the role of T cells and their activation as a potential clinical driver in autoinflammation of unknown origin, despite the common assumption that autoinflammation is predominantly associated with innate immunity dysregulation. An additional intriguing finding from our study is the overlap in immune signatures between FMF and Behçet's disease, where the discrimination between these two conditions were the most challenging. Notably, clinical and genetic overlap has also been observed in both diseases, with pathological variants of *MEFV* identified in both diseases, further reinforcing the potential connection between them[41].

There are several key limitations to this study. Although multicenter, the study needs to be independently verified to determine if the changes observed are reproducible across multiple cohorts. The phenotypes observed here in these patients likely reflect different stages of disease as they were at varying points in their disease onset; there is potential for an immunological evolution during disease progression. Diagnostic criteria for SAIDs are notoriously difficult to implement, and some patients defy simple categorization, impeding simple replication studies. While the study was powered to detect large effect sizes, smaller effect sizes will have been missed (Supplementary Table 1). Finally, flow cytometry and serum proteomics do not measure RNA expression, and thus, intracellular changes reporting on key mechanistic pathways will have been missed. However, even noting these limitations, the parallels between autoinflammation of unknown origin and Still's disease provide actionable research pathways, and the overall finding of similarity was consistent across both the flow cytometric approach and the plasma mass spectrometry approach. The phenotypes observed here could be incorporated into biomarker studies to aid in diagnostics. Genetic similarity could also be tested, using known Still's disease genetic drivers as candidate genes in autoinflammation of unknown origin. Finally, immunophenocopying could potentially be accompanied by parallel treatment response. Still's disease is typically treated with corticosteroids, methotrexate, IL1 blockade and anti-IL6 biotherapies. The study here, following replication, might provide a rationale for testing effective Still's disease regimes in patients with severe autoinflammation of an unknown origin as no specific drug is currently recommended, and early treatment is essential for preventing complications such as amyloidosis and further organ damages. The beneficial response to IL1 blockade of the majority of tested patients with undifferentiated systemic autoinflammatory disorder[42–44], as observed in Still's disease, provides further support for exploring this approach.

## Methods

### Study design and patient's characteristics

The study was designed to investigate immune signatures and to define biomarkers specific to different autoinflammatory disorders, in order to allow better disease stratification and faster, as well as more precise diagnosis. This study was conducted within the Immunome project consortium for AutoInflammatory Disorders (ImmunAID). 450 patients with different systemic autoinflammatory disorders (SAID) and healthy controls were enrolled from 30 clinical centers throughout Europe and Turkey, to assure a broad inclusion of SAID subtypes

and a sufficient number of individuals for rare diseases. For Still's disease[45,46], Behçet's disease[47,48] and Familial Mediterranean Fever (FMF)[49], existing and validated international classification criteria were used. For autoinflammation of an unknown origin, without existing classification criteria, diagnosis was made according to published definitions of "undifferentiated" or "undefined" SAID[2,26,27]. Patients were selected with idiopathic and recurrent systemic inflammation according these following criteria: 1) Temperature >38 °C during flare, CRP > 30 mg/L or SAA > 25 mg/L on at least 3 occasions separated by 2 weeks and symptom occurring over a period of more than 3 months, and 2) no alternative SAID (or any other disease) diagnosis, during a year-long follow-up period. Furthermore, for each condition, specific inclusion criteria for active disease were defined and applied: For Still's: disease: fever and CRP ≥ 3 ULN and 2 of the following features: arthritis, sore throat, skin rash, lymphadenopathy, increased WBC ≥ 10.000/mm3, PMN > 80%. For Behçet: Physician Global Assessment (PhGA) of clinical symptoms ≥ 2/10, parent/patient global assessment ≥ 2/10, fever and/or CRP ≥ 3 ULN. For FMF: fever, PhGA ≥ 2/10, Patient AIDAI score ≥ 9 on an at least 29 day period, CRP ≥ ULN (or ≥ 10 mg/l) or/and SAA ≥ 10 mg/L. For Autoinflammation of unknown origin: fever, CRP ≥ 30 mgL, PhGA ≥ 2/10 (Supplementary Spreadsheet 1). Exclusion criteria included infection, antibiotic treatment in the last 2 weeks, autoimmune diseases, evidence of immunodeficiency or neoplasia. All Autoinflammation of unknown origin patients failed to meet the diagnostic criteria of any alternative SAID, both at inclusion and during a one-year follow-up period. None of these patients presented with the characteristic Still's disease features of a quotidian spiking fever or transient salmon-pink or evanescent erythematous rash, and all demonstrated no mutations in the VEXAS syndrome-associated UBA1 gene. Healthy individuals (free of inflammatory disorders) were used as a control group regarding baseline immune parameters. Demographics of the healthy and patient populations are given in Table 2. When possible, patient samples were collected prior to treatment; when patients were already undergoing treatment the treatments were recorded (Supplementary Spreadsheet 1). From the ImmunAID cohort, 52 patients were diagnosed with Still's disease in adults, 34 with Behçet disease, 49 with FMF, 71 with autoinflammation of unknown origin and an additional 72 healthy controls were included. After exclusion of patients without available samples, and patients with samples of poor quality to perform further analysis, we enrolled in our study 34 patients diagnosed with Still's disease in adults, 23 with Behçet disease, 35 with FMF, 36 with autoinflammation of unknown origin and 58 healthy controls. All patients meeting the inclusion criteria and not excluded for the defined reasons above are included in the analysis, providing a representation of patients of both sexes (as self-reported). Disaggregated data for sex is available in Supplementary Spreadsheet 3 and 4, and post-hoc sex-based comparisons were not performed due to limiting statistical power.

All individuals provided written informed consent for data collection, sampling and analyses. Consenting committees that approved the study were:

Ethics Committee of 14 French clinical sites: Comité de Protection des Personnes Ouest 6 (EC Number: [19.01.03.64304], Approval Date: [08/04/2019]);

Ethics Committee: hacettepe university clinical research ethics boards (EC Number: [2020/08-09 (KA-20034)], Approval Date: [02/06/2020]);

Ethics Committee: Commission Cantonale d'Ethique de la Recherche (EC Number: [2020-01334], Approval Date: [02/11/2020]);

Ethics Committee: CEIm Fundació Sant Joan de Déu (EC Number: [PIC-112-19], Approval Date: [05/07/2019];

Ethics Committee: Republic of Slovenia National Medical Ethics Committee (EC Number: [0120-495/2019/4], Approval Date: [15/11/2019]);

Ethics Committee: Medische Ethische Toetsings Commissie (EC Number: WT/ss/MEC-2020-0164], Approval Date: [112/10/2020]);

Ethics Committee: General hospital of Athens's board of directors (EC Number: [969/12-10-2020], Approval Date: [13/11/2020]);

Ethics Committee: Research UZ/KU Leuven (EC Number: [S63731], Approval Date: [12/08/2020]);

Ethics Committee: OPBG's Ethics Commitee (EC Number: [1357], Approval Date: [24/11/2020]);

Ethics Committee: Comitato Etico Milano Area 1 (EC Number: [2022/EM/060], Approval Date: [16/03/2022]);

Ethics Committee: Comitato Etico Regionale per la Sperimentazione Clinica della Regione Toscana Sezione: AREA VASTA SUD EST (EC Number: [19577], Approval Date: [21/02/2022]);

Ethics Committee: Comitato Etico per la Sperimentazione Clinica della Provincia di Padova (EC Number: [5350/AO/22], Approval Date: [10/03/2022]);

Ethics Committee: Medical Association of Westphalia-Lippe and the Westfälische Wilhelms-University of Münster (EC Number: [2020-476-f-S], Approval Date: [15/12/2020]);

Ethics Committee: South West - Frenchay Research Ethics Committee (EC Number: [20/SW/0022276218], Approval Date: [18/08/2020])

Sample collection was performed according to the local ethical requirements. Participants had blood drawn to collect peripheral blood mononuclear cells on site. Samples were frozen and locally stored in liquid nitrogen. Samples were shipped to a central laboratory in batches and stored in liquid nitrogen until further analysis.

### PBMC isolation and flow cytometry
Peripheral blood mononuclear cells (PBMC) were isolated from EDTA whole blood by density gradient centrifugation, using LSM (lymphocyte separation medium, MP Biomedicals, 0850494) according to the

## Table 2 | Study population demographics

| | N | Sex | | | Age (years) | | | | | CRP level (mg/L)† | | | |
|---|---|---|---|---|---|---|---|---|---|---|---|---|---|
| | | N (%) female | P- value* | Median | IQR | Min | Max | P- value^ | Median | IQR | Min | Max |
| Healthy | 58 | 31 (53.4%) | – | 30 | 13.8 | 10 | 59 | – | – | – | – | – |
| Inflammation of unknown origin | 36 | 17 (47.2%) | 0.666 | 44.5 | 32 | 1 | 76 | 0.425 | 62.5 | 66.1 | 1 | 342 |
| Still's disease | 34 | 24 (71.4%) | 0.118 | 30 | 12.5 | 16 | 66 | 0.855 | 90.8 | 100 | 6.3 | 374 |
| Familial Mediterranean Fever (FMF) | 35 | 23 (65.7%) | 0.206 | 26 | 24 | 3 | 74 | 0.321 | 13.8 | 25.2 | 1 | 293 |
| Behçet's disease | 23 | 9 (39.1%) | 0.335 | 23 | 23 | 6 | 58 | 0.126 | 20.4 | 26 | 1.4 | 286 |

IQR, interquartile range. * P-value, chi-squared test of sex in relation to disease group compared to healthy. ^ P-value, Mann-Whitney U test (two sided) for age in relation to disease group, compared to healthy. † P-value, Kruskal-Wallis test (two sided) for CRP levels among disease groups = 0.736.

manufacturer's recommendations, frozen in Cryostor CS10 (Stemcell technologies, 07959) and stored in liquid nitrogen. Frozen PBMCs were thawed in complete RPMI, washed twice, and stained with live/dead marker (fixable viability dye eFluor780, eBioscience 65-0865-14) in the presence of an Fc receptor binding inhibitor cocktail (eBioscience, 14-9161-73). Cells were fixed using the Foxp3/Transcription factor buffer staining set (eBioscience, 00-5523-00), stained with fluorochrome-conjugated antibodies (Supplementary Table 2), and data were collected on a BD FACSymphony A3 Cell Analyzer (BD Biosciences). Data from.fcs files were compensated with AutoSpill [83] and manually pre-processed in FlowJo to exclude non-cellular events. Gating was performed in R (version 4.0.2) with a custom script (Neumann et al., manuscript in preparation), automatically calculating subset boundaries between populations listed in Supplementary Spreadsheet 2.

## Flow cytometry analysis

The custom script for automatic gating was set to identify 261 different cell populations or subpopulations with distinct phenotypes. Following the automated gating (Supplementary Fig. 12), a manual curation of the data was conducted, which found variation in CD80 marker detection resulted in low identification capacity, resulting in exclusion from the analysis. Variation in low-frequency subsets made the calculation of sub-populations from these subsets difficult, due to missing data. To maintain comparability between individuals and consistency within the database, we excluded cell populations with more than 20% missing data. Consequently, data analysis was performed on a set of 208 immune variables (cell population frequencies), available as Supplementary Spreadsheet 3.

To address missing data, data was imputed, resulting in a reduction in data precision rather than data loss. K-nearest neighbors (KNN) imputation[50] was used to maintain data heterogeneity and provide good accuracy. Unlike imputation methods based on central tendency and regression models, KNN imputation does not significantly reduce data heterogeneity. Each cellular variant frequency exhibited its own unique distribution and magnitude. To prevent highly skewed distributions and large values from disproportionately influencing the distance measures, a Box-Cox transformation[51] was applied, followed by standardization prior to imputation. This process normalized the data, ensuring that the resulting dataset contained information more comparable in their metrics, for a more robust and consistent database for subsequent analyses.

## Mass spectrometry

Peripheral blood was collected into 7 mL EDTA-coated tubes and processed within 2 hours of collection. Blood samples were first gently inverted 5–6 times to ensure proper mixing with the anticoagulant. The tubes were centrifuged at $400 \times g$ for 5 min at room temperature. After centrifugation, the plasma fraction was carefully aliquoted into 2 mL cryogenic tubes (Nalgene, 028038), avoiding disturbance of the buffy coat layer. Collected plasmas were stored at $-80\,°C$ until further analysis.

Plasma samples were analyzed, together with quality-control samples, which consists of pooled plasma provided by healthy individuals (two women and one man), processed under the same conditions as those described for other plasma samples. Each biological sample was analyzed once by LC–MS/MS (no technical LC–MS replicates). Samples were processed in randomized batches of 18 patient samples, along with two times the processed quality control sample, which were processed under the same conditions and in parallel with the patient samples. Each batch of 20 samples was prepared in parallel and injected in a single injection series, in the following order: QC1, patient samples 1–9, QC2, patient samples 10–18.

The protein content of each plasma was quantified using RCDC quantification kit (Biorad, 5000122) to load a constant amount of protein (600 µg) on depletion columns (High Select™ Top14 Abundant Protein Depletion Mini Spin Columns Ref A36370, ThermoFisher Scientific), which removed the 14 most abundant proteins in the plasma, such as the Albumin, IgA, IgD, IgE, IgG, IgM, IgG (light chains), alpha1-acid glycoprotein, fibrinogen, haptoglobin, alpha1-antitrypsin, alpha2-macroglobulin, transferrin, and the apolipoprotein A-1. After, we concentrated the eluate from the depletion column using an Amicon Ultra 3 kDa cut-off (Millipore, UFC500396). Depleted plasma protein content was then quantified using the microBCA kit (Pierce, Thermo-Fisher Scientific, 23235) and an aliquot of 10 µg was reduced using dithiothreitol 10 mM with incubation at 56 °C for 40 min (Thermo-Fisher Scientific, J15397), alkylated using iodoacetamide 20 mM with incubation at room temperature for 30 minutes (BioUltra, Sigma-Aldrich, Merck, I1149). The excess of idioacetamide was removed by adding 11 mM dithiothreitol (incubation 10 minutes at room temperature), followed by a digestion using Trypsin/LysC (Promega, V5073) at a 1/25 enzyme/protein ratio for 3 h at 37 °C. A second addition of trypsin/LysC at a ratio of 1/50 was performed, and the samples were incubated overnight at 37 °C. Digestion was stopped by adding trifluoroacetic acid to a final concentration of 0.5%. An aliquot of 3 µg was then purified on C18 tips (Pierce, ThermoFisher Scientific, 87782) and dried under vacuum. Peptides were resolubilized in 100 mM ammonium formate (pH 10), and a volume corresponding to 2 µg of peptides was injected onto 2D-nanoAcquity UPLC (Waters, Corp., Milford, USA) coupled online with a Q Exactive Plus (ThermoFisher, USA, IQLAAEGAAPFALGMBDK). Each sample was spiked with a commercial mixture of protein digest standards originating from non-human biological material: the MassPREP™ Digestion Standards (Waters, Corp., Milford, USA, 186002865, 186002866), at 150 fmol of ADH per sample. This commercial standard consists of two standard mixtures (MPDS Mix 1 and MPDS Mix 2) containing protein digests of Yeast Alcohol Dehydrogenase (ADH), Rabbit Glycogen Phosphorylase b, Bovine Serum Albumin, and Yeast Enolase, all present at a known protein ratio, allowing for the relative quantitation of the spiked samples. The samples were loaded at 1.5 µL/min [20 mM ammonium formate solution (Merck NH4OH 1.05432.1011 with Biosolve Formic Acid 00069141A8BS) adjusted to pH 10] onto the Peptide BEH C18 column (Waters, Corp., Milford, USA, nanoEase M/Z Peptide BEH C18, 186009268) and eluted in three fractions, with a flow rate of 1.5 µL/min with 13.3, 19 and 65% acetonitrile (Biosolve Chimie, Dieuze, France, ULC/MS-CC/SFC 0001204102BS) in the mobile phase onto the low pH columns. Each fraction after a ten times online dilution to pH 3 was loaded onto the Symmetry trap column (Waters, Corp., Milford, USA, nanoEase M/Z Symmetry C18, 186008821) (low pH pump 15 µL/min, 99.9/0.1 A/B solvent) and subsequently separated on the Analytical column (Waters, Corp., Milford, USA, nanoEase M/Z HSS C18 T3, 18608818) at a flow rate of 250 nL/min and temperature of 40 °C with solvent A (0.1% formic acid (Biosolve Chimie, Dieuze, France, ULC/MS-CC/SFC 00069141A8BS) in water) and solvent B (0.1% formic acid (Biosolve) in acetonitrile (Biosolve))as follows: 1 minute at constant solvent 99% A then linear gradients reaching 7% solvent B at 5 minutes and 35% solvent B at 140 minutes. The total runtime, including wash and reconditioning of the column for each fraction, was 180 min. The mass spectrometer method was a Top12-MSMS method (singly charged precursors excluded). The parameters for MS spectrum acquisition were as follows: mass range, 400 to 1750 m/z; resolution, 70,000; AGC target, $1 \times 10^6$; or maximum injection time, 200 ms. The parameters for MS2 spectrum acquisition were as follows: an isolation window of 2.0 m/z, a normalized collision energy (NCE) of 25, a resolution of 17,500, an AGC target of $1 \times 10^5$ or a maximum injection time of 50 ms, and an underfill ratio of 1.0%. The main parameters for Q Exactive Plus tune with NanoESI source were: spray voltage of 2.2 kV, capillary temperature of 270 °C, and S-Lens RF level of 50.0.

The raw files of 447 patients from the ImmunAID cohort and the 50 QC samples were normalized with the label-free quantification

(LFQ), using FragPipe (v22.0) for the analysis of raw files in DDA mode (available as Supplementary Spreadsheet 4)[52–54]. MS/MS spectra were analyzed using MSFragger (v4.1) search engine with the following settings: the SwissProt database was restricted to the human taxonomy (downloaded on 31st of January 2021, 20394 sequences. The four proteins of the Mass Prep Digestion Standard and porcine trypsin sequences were added to the FASTA file for interrogation). FragPipe was used to generate decoys; the final database contains 40888 entries (20444 decoys: 50.0%). Oxidation of methionine (M) and deamidation of Asparagine or Glutamine (N, Q) were set as variable modifications, while carbamidomethylation of the cysteines was set as a fixed modification. The maximum number of missed cleavages was set at two using strict trypsin as the enzyme (cleavages at K and R sites, even if followed by a proline), and the minimal peptide length for identification was set at 7 amino acids. The precursor mass tolerance was set at 4.5 ppm (4.5 ppm upper and − 4.5 ppm lower limits), and the fragment mass tolerance was set at 20 ppm. The protein false discovery rate (FDR) was set to 5% (FragPipe validation step). Data normalization was performed using the MaxLFQ algorithm (IonQuant v1.10.27)[55]. The minimum ions count for LFQ was set at two. Match between runs (MBR) with an ion FDR of 0.01 and normalization across runs was used on razor and unique peptides with a parameter of 19 MBR top runs. LFQ normalized data were log2 transformed, and proteins with more than 15% missing values across all disease groups were excluded prior to downstream analyses. Finally, patients were matched to the flow cytometry dataset, leading to diseases groups containing 36 patients with autoinflammation of unknown origin, 34 patients with Still's disease, 35 patients with FMF, and 23 patients with Behçet, and 57 healthy controls.

## Plasma analysis

Plasma ferritin levels were quantified by Chemiluminescent Microparticle ImmunoAssay (CMIA) using the Alinity i Ferritin Reagent Kit (Abbott CE-IVD, Product Number 07P65) and the Alinity i analyzer (Abbott Laboratories, Abbott Park, IL, USA), following the manufacturer's instructions. Briefly, samples were analyzed either undiluted or, if values exceeded the linear range (up to 1675.6 ng/mL), diluted in the range of 1:1-1:20. The assay involved an automated two-step sandwich. First, immunoassay samples were incubated with paramagnetic microparticles coated with mouse monoclonal anti-ferritin antibodies, washed, and then incubated with an acridinium-labeled rabbit polyclonal anti-ferritin conjugate. Secondly, following a second wash, pre-trigger and trigger solutions were added, and the resulting chemiluminescent signal (relative light units) was measured by the optical system. Ferritin concentrations were determined by the system software using a stored 4-parameter logistic calibration curve generated with the Alinity i Ferritin Calibrators, reported in ng/mL within a measuring interval of 1.98–1675.6 ng/mL. All control measurements were performed in accordance with the kit's IVD certification to meet the required analytical performance criteria.

## Identification of parameters associated with disease

For independent analysis of immune parameter association, we employed multivariate logistic regression for each immune parameter, using sex and age as confounding variables. The outcome variable in these regressions was the presence of each pathology compared to healthy controls, allowing the isolation of the specific contribution of each cellular subset to the disease state.

To select the most associated markers, first all markers with a p-value less than or equal to 0.05 were identified. Among these significant markers, markers were ranked based on the magnitude of the effect, using the odds ratio (OR) as the criterion. The optimal number of markers to include in the model was assessed using multivariate logistic regression with L2 penalization (ridge regression). This penalization reduces numerical instability and limits potential overfitting. The model was run with 1 to 200 variants together, in order of importance. For each of these 200 models, executed randomly 200 times using 5-fold cross-validation, the outcome was stratified to capture the lower number of immunological variables with the highest explainability and generalization.

The optimal number of markers necessary to explain each disease was identified by analyzing the behavior of the average area under the ROC curve (AUC) across the validation groups for each number of variants.

## Describing similarities between different pathologies

To visualize the similarities between different pathologies in relation to the control group, principal component analysis (PCA) focusing on the first two principal components was used. PCA analysis was run both on all cell frequencies in the dataset, where natural variation dominated, and after selecting the n most informative variables based on the optimal number of variables that differentiated the pathologies from the control group. This selection process was based on the strength of association with the diseases.

## Distinguishing different patient phenotypes based on immune markers

To estimate the effect of immune markers in discrimination across diseases, we chose to use Random Forest (RF) due to its random selection of predictors evaluated in each tree, which minimizes the impact of multicollinearity among markers, providing a broader view of potentially relevant markers. We assessed each combination of patient phenotypes using stratified 10-fold cross-validation. In each fold of the training group, we ran RF models with 5001 trees and varying maximum tree depths to regulate model generalization. The model with the best out-of-bag estimations was selected to construct the ROC curve for each fold. Finally, the 10 ROC curves were interpolated to obtain a mean curve and a 95% confidence interval. As parameter estimations were performed within each fold, there was no need for an additional evaluation group, which would be problematic given the high dimensionality and limited number of instances in the dataset. Excluding healthy individuals, we used the remaining data to train a Random Forest model with 5001 trees and a maximum tree depth limited to 8. This model was used to estimate impurity-based feature importance as described[56].

## Data availability

All raw data, flow cytometry and proteomics, generated by members of the ImmunAID consortium, are stored in the European Genome–Phenome Archive (EGA) under accession number EGAS50000001393 (https://ega-archive.org/studies/EGAS50000001393 and the Dataset ID: ICKD2021572). Access to these data is controlled because they originate from human participants and may be potentially re-identifiable when combined, and their use is restricted by institutional ethics approval and applicable data protection regulations. Researchers who wish to access the data may submit a request to the corresponding Data Access Committee via the EGA portal; requests will be evaluated by the ImmunAID data managers to ensure that the proposed use complies with the original informed consent and regulatory constraints. Individual de-identified participant data is available for sharing, with the raw data files and clinical summaries. Extended minimum datasets, including demographic data, linked clinical summaries and input flow cytometry and proteomics data for analysis are available as Supplementary Spreadsheets 3 and 4. The data analysis for this paper can be recreated using the extended minimum datasets and the source code. Source data from the output of this analysis are provided with this paper. Source data are provided in this paper.

## Code availability

Source code for the analysis is available on GitHub at: https://github.com/rafael-veiga/ImmunAID release v1.2.0 (https://doi.org/10.5281/zenodo.18436096). FragPipe parameters are available at https://gitlab.uliege.be/giga-rheumatology/public/immunaid-ms-parameters (https://doi.org/10.5281/zenodo.18413673).

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

## Acknowledgements

The authors thank all patients and volunteers enrolled in the study. This work has received funding from the European Union's Horizon 2020 research and innovation program under grant agreement No 874707 (EXIMIOUS), and No 779295 (ImmunAID) to A.L., S.H.B., and D.D.S. The authors acknowledge the important contributions of Jeason Haughton, the KU Leuven FACS Core, and the GIGA Proteomics Facility.

## Author contributions

R.V. led the statistical analysis and data display. L.D.V. led on flow cytometry sample analysis. C.P. led on mass spectrometry sample analysis. R.V., L.D.V. and C.P. are equal contribution first authors. J.N., L.B., T.P., M.W., M.F., G.C., D.B. and G.B. provided expertise in data generation. C.W. led on clinical analysis. S.V., E.C., B.F. and C.W. led on clinical sample collection and were involved in clinical analysis, on behalf of the ImmunAID consortium. P.M., C.W., D.D.S., S.H.B. and A.L. designed the study. D.D.S., S.H.B. and A.L. supervised the execution of the research. A.L. wrote the initial draft of the manuscript. C.W., D.D.S., S.H.B. and A.L. are equal contribution last authors. All authors contributed to the final manuscript.

## Competing interests

B.F. has received consulting fees from Novartis and SOBI. The remaining authors declare no competing of interest.
