## [Transparent Peer Review file · Nature Communications]

Adult patients with autoinflammation of unknown origin partially phenocopy the immune presentation of Still's disease

Corresponding Author: Professor Adrian Liston

Version 0:

Reviewer comments:

Reviewer #1

(Remarks to the Author)

The authors collected PBMCs from treatment-naive and active SAID patients (n=36), healthy individuals (n=58), and patients with other inflammatory diseases (AOSD, Behçet's disease, FMF, total n=93) to analyze and compare altered cell subsets using flow cytometry. They observed an increased number of CD38-positive cells in SAID and, through machine learning and other methods, found similarities and differences between SAID and other groups, concluding that the patterns of SAID and AOSD were similar.

SAID is a critical issue in real-world clinical practice, making it essential to focus on these patient populations to drive research. However, rigorous examination of symptom trends and treatment approaches is necessary. This study lacks an analysis of the clinical aspects of the disease, relying solely on quantitative comparisons of specific leukocyte subsets in peripheral blood. These parameters alone are insufficient for discussing differences between SAID and other diseases. Integrating additional analyses, such as proteomics and single-cell profiling, while focusing on specific clinical manifestations overlapping with other autoinflammatory diseases, would be more informative (e.g. FMF vs. FMF-like autoinflammatory disease).

The definition of SAID provided in the Methods section (p. 11) is vague and overly heterogeneous, describing the condition only in terms of fever, inflammatory responses, and some persistent symptoms. There is no information regarding precise treatments and symptoms at the time of sample collection for patients with control diseases. Furthermore, defining periodic fever over a duration of 3 months for SIAD is weak and raises questions about its validity as a disease concept. A strict consensus on the detailed clinical manifestations of SAID is required. Employing a more rigorous definition and conducting cluster analyses based on various parameters would allow for a clearer characterization of the patient population.

While deep phenotyping is mentioned in the abstract, the analysis is cross-sectional and conducted at a single point in time, lacking exploration of relationships between disease status, laboratory findings such as C-reactive protein levels, and treatment changes. The term "treatment naive" is ambiguous—does it imply that no treatments, including glucocorticoids or NSAIDs, were ever administered? Additionally, it is unclear whether the disease control population was also treatment naive. These descriptions are insufficiently detailed and require clarification.

(Remarks on code availability)

Reviewer #2

(Remarks to the Author)

The paper is pretty interesting and deserves consideration. It is well suitable for the Journal and brings to a significant contribution to the state-of-the-art.

I would like to add some remarks and suggestions to further improve the quality of the paper, as reported below:

- In the Introduction, it would be beneficial to also include some epidemiological features of SAIDs, including their prevalence over the globe or in some specific areas. In addition, a table resuming the main clinical hallmarks related to

SAIDs, could improve the readability of the paper at a glance.

- For sample selection, apart from the details correctly mentioned in the paper, did you perform an eventual power analysis or something similar, that could drive to a quantitative assessment of the impact of your investigation?

- It is necessary to acknowledge the ethical approval(s) of the protocol followed by means of the number and name of the approving authority.

- As for the missing data management, how did you perform the imputation of data?

(Remarks on code availability)

Version 1:

Reviewer comments:

Reviewer #1

(Remarks to the Author)

The authors have performed additional analyses in response to the previous review, which have substantially strengthened their dataset.

Major comments

1. Using cell-surface markers and newly performed proteomic profiling, the authors demonstrate similarities between autoinflammation of unknown origin (AUO) and Still's disease. However, could this AUO cohort in fact represent patients who should have been classified as having Still's disease? Although the Supplementary Table lists the AUO patients and their clinical features, the authors should specifically extract key Still's disease-related manifestations (e.g. high-grade periodic fever, salmon-pink rash, as defined in the ILAR criteria), together with distinctive features (e.g. urticaria, ulcers), present a focused comparison of these features, and clearly demonstrate that the AUO patients are indeed distinct from those with Still's disease.

2. The authors newly performed proteomic analyses; did classical Still's disease-related markers such as ferritin, IL-18, CD168, and CXCL9 differ between the AUO and Still's disease groups?

Minor comments

3. In Figures 1F and 2C, the arrows overlap with the text, making these panels difficult to read. The layout should be adjusted to avoid this overlap.

4. In Figure 2A, NK- and B-cell levels appear lower in the untreated group than in the treated group. The relevant significance levels (e.g. p-values) should be indicated. Given the large number of figures, the overall presentation is hard to follow; the authors should consider reducing the number of main figures and moving some panels to the Supplementary Material.

5. For Figures 1, 2, 4, 5, and 6, the corrected statistical significance levels, adjusted for the number of tests (multiple comparisons), should be indicated directly on the plots.

6. For Figure 3, the current presentation of all plots and ROC curves is difficult to interpret. It would be clearer if the authors could summarise these data in a table.

7. In the Limitations section, the authors should also acknowledge that they did not assess RNA expression, and therefore the intracellular mechanisms underlying these findings remain insufficiently understood.

8. Some of the elderly male AUO patients may in fact have VEXAS syndrome. Have the authors performed somatic UBA1 genotyping in this cohort? If not, this should be mentioned in the discussion.

(Remarks on code availability)

Reviewer #2

(Remarks to the Author)

I warmly thank the authors for having carefully and positively considered my suggestions. In my opinion, the manuscript is now more complete and its quality is higher than in the previous round. I just have a very minor remark, considering the thorough paragraph added about epidemiological features. In this regard, I would appreciate the inclusion of some bibliography to support the numbers included.

(Remarks on code availability)

Version 2:

Reviewer comments:

Reviewer #1

(Remarks to the Author)
No additional comments.

(Remarks on code availability)
No comments.

Reviewer #2

(Remarks to the Author)
All my concerns were successfully answered. Thank you.

(Remarks on code availability)

Point-by-point response to reviewers

We thank both reviewers for their considered and expert evaluations. The points which they have raised have brought about major improvements to our study. We thank the reviewers for their patience in this second submission, as the patient-by-patient clinical evaluation and new mass spectrometry data generation took a substantial effort by large teams. We do agree that this was worth the effort, as the study is now far more valuable to the community.

Reviewer #1

The authors collected PBMCs from treatment-naive and active SAID patients (n=36), healthy individuals (n=58), and patients with other inflammatory diseases (AOSD, Behçet's disease, FMF, total n=93) to analyze and compare altered cell subsets using flow cytometry. They observed an increased number of CD38-positive cells in SAID and, through machine learning and other methods, found similarities and differences between SAID and other groups, concluding that the patterns of SAID and AOSD were similar.

SAID is a critical issue in real-world clinical practice, making it essential to focus on these patient populations to drive research. However, rigorous examination of symptom trends and treatment approaches is necessary. This study lacks an analysis of the clinical aspects of the disease, relying solely on quantitative comparisons of specific leukocyte subsets in peripheral blood. These parameters alone are insufficient for discussing differences between SAID and other diseases. Integrating additional analyses, such as proteomics and single-cell profiling, while focusing on specific clinical manifestations overlapping with other autoinflammatory diseases, would be more informative (e.g. FMF vs. FMF-like autoinflammatory disease).

We thank the review for focusing on the real-world importance of understanding the phenotype of patients with autoinflammation of unknown origin. Our study is valuable to the community precisely because we have a large well-characterised cohort, with relevant real-world disease controls, in addition to highly detailed biomedical parameters.

In response to the reviewer, we have:

1) Performed an additional unbiased proteomics analysis using mass spectrometry on the plasma of each patient, doubling the amount of presented data in the manuscript. The core findings of the original manuscript, notably the closest similarity between Still's disease and autoinflammation of unknown origin, is replicated in this new dataset, using entirely independent biomedical read-outs.

2) Included a detailed clinical description of the patients in Supplementary Spreadsheet 1, covering the inclusion read-outs at evaluation point and sample collection point, and a detailed list of musculoskeletal manifestations, mucocutaneous involvement, lesions, ear-nose-throat manifestations and gastrointestinal-liver manifestations. This provides the specialist reader with an understanding of the symptoms on a patient-to-patient basis. Analogous information is also provided for all control patients. We also provide corresponding data per patient in Supplementary Spreadsheet 3, which, for the first time, allows independent research groups to make their own independent sub-analysis, for example looking at similarity between patients with and without gastrointestinal involvement across the patient groups. These comparisons are beyond the scope of the paper to present

here, however the paper now includes all the resources required for any such comparison.

The definition of SAID provided in the Methods section (p. 11) is vague and overly heterogeneous, describing the condition only in terms of fever, inflammatory responses, and some persistent symptoms. There is no information regarding precise treatments and symptoms at the time of sample collection for patients with control diseases. Furthermore, defining periodic fever over a duration of 3 months for SIAD is weak and raises questions about its validity as a disease concept. A strict consensus on the detailed clinical manifestations of SAID is required. Employing a more rigorous definition and conducting cluster analyses based on various parameters would allow for a clearer characterization of the patient population.

We thank the reviewer for this point. The ImmunAID consortium included a major effort to harmonise diagnosis of patients, by applying consensus validated classification criteria for all SAIDs included in this study. Patient entry was based on recurrent fevers and increased acute phase reactants (CRP >30mg/L or SAA>25mg/L) over a period of 3 months. For most conditions here, the precise SAID of diagnosis was then based on additional diagnostic inclusion criteria, while for autoinflammation of unknown origin it was diagnosis of disease exclusion. Patients remained as “autoinflammation of unknown origin” if they failed to meet the diagnostic criteria of any alternative SAID, during a year-long follow-up period. This has now been clarified in the text:

“For Still’s disease ^{35, 36}, Behçet’s disease ^{37, 38} and Familial Mediterranean Fever (FMF) ³⁹, existing and validated international classification criteria were used. For autoinflammation of an unknown origin, without existing classification criteria, diagnosis was made according to published definitions of “undifferentiated” or “undefined” SAID ^{2, 16, 17}. Patients were selected with idiopathic and recurrent systemic inflammation according these following criteria: 1) Temperature >38°C during flare, CRP> 30mg/L or SAA>25mg/L on at least 3 occasions separated by 2 weeks and symptom occurring over a period of more than 3 months, and 2) no alternative SAID (or any other disease) diagnosis, during a year-long follow-up period. Furthermore, for each condition, specific inclusion criteria for active disease were defined and applied: For Still’s: disease: fever and CRP ≥ 3 ULN and 2 of the following features : arthritis, sore throat, skin rash, lymphadenopathy, increased WBC ≥ 10.000/mm³, PMN > 80%. For Behçet: Physician Global Assessment (PhGA) of clinical symptoms ≥ 2/10, parent/patient global assessment ≥ 2/10, fever and/or CRP ≥ 3 ULN. For FMF: fever, PhGA ≥ 2/10, Patient AIDAI score ≥ 9 on an at least 29 day period, CRP ≥ ULN (or ≥ 10 mg/l) or/and SAA ≥ 10 mg/L. For Autoinflammation unknown origin: fever, CRP ≥ 30 mg/L, PhGA ≥ 2/10. Exclusion criteria included infection, antibiotic treatment in the last 2 weeks, autoimmune diseases, evidence of immunodeficiency or neoplasia.”

In the revised paper we have also included a new resource, Supplementary Spreadsheet 1, which goes in detail through the mucocutaneous, musculoskeletal, gastrointestinal, liver, ear-nose-throat manifestations, and information on fever, for each patient. This data is further linked to the flow cytometry data, providing a unique resource to any researchers seeking to perform their own analysis linking phenotype to clinical symptom. The precise treatment exclusion criteria are now also given in the methods.

While deep phenotyping is mentioned in the abstract, the analysis is cross-sectional and conducted at a single point in time, lacking exploration of relationships between disease status, laboratory findings such as C-reactive protein levels, and treatment changes.

In the revised manuscript we have substantially increased the clinical phenotyping, covering multi-systems manifestations and disease activity status in each system and with regards to CRP. We also detail treatment information for each patient. We have also double the biomedical readouts, including a completely new set of plasma mass spectrometry proteomics data. While we have not performed every possible subset analysis with this data, the open resource nature of our paper allows researchers to perform this analysis independently. The reviewer correctly raises the issue that the study is cross-sectional in design: in practice, this cohort took >5 years to collect the samples for, and longitudinal analysis was unfeasible. It is, however, already the largest systems immunology analysis ever released for this under-studied patient group. We raise the lack of longitudinal data as a limitation in the discussion of our paper.

The term "treatment naive" is ambiguous—does it imply that no treatments, including glucocorticoids or NSAIDs, were ever administered? Additionally, it is unclear whether the disease control population was also treatment naive. These descriptions are insufficiently detailed and require clarification.

We agree with the reviewer, and have extensively clarified this issue. We now no longer use the term "treatment naïve" and instead use "untreated" and "treated". "Untreated" refers to all prior prescribed treatments, including glucocorticoids and NSAIDs, while treated refers to any prescribed treatments. This is clarified in the methods, and the data on the treatment per patient is given in Supplementary Spreadsheet 1. We also provide a new analysis of treated vs untreated patients in Figure 2 and Figure 6, which demonstrates that the immune phenotypes observed (cellular and proteomic, respectively) are driven by disease status and not treatment status.

Reviewer #2

The paper is pretty interesting and deserves consideration. It is well suitable for the Journal and brings to a significant contribution to the state-of-the-art.

We thank the reviewer for their thoughtful and positive evaluation.

I would like to add some remarks and suggestions to further improve the quality of the paper, as reported below:

- In the Introduction, it would be beneficial to also include some epidemiological features of SAIDs, including their prevalence over the globe or in some specific areas. In addition, a table resuming the main clinical hallmarks related to SAIDs, could improve the readability of the paper at a glance.

We agree, and have added the following paragraph:

"The prevalence of SAIDs ranges from 1 in 1000 to 1 in 1,000,000 people, depending on the specific disease, country, and population. Still's disease (encompassing both Adult-onset Still's Disease and systemic Juvenile Idiopathic Arthritis) presents a relatively uniform global distribution with an estimated prevalence of 1-10 cases and 10-100 cases per 1,000,000

population, for the adult and juvenile-onset form respectively, and occurring at similar rates across diverse populations worldwide. Familial Mediterranean Fever exhibits a pronounced ethnic and geographic concentration, with a significantly higher prevalence among specific Mediterranean and Middle Eastern populations. Prevalences vary between 1/250 (Sephardic Jews) and 1/2600 (Arabs). (1:1,000), and Arabs (1:2,600). Significant prevalence rates are seen in countries like Greece, Italy, Spain, and Portugal, as well as in diaspora communities where affected populations have migrated. Behçet's disease demonstrates a geographic pattern following the ancient Silk Road trading route, with Turkey showing the highest global prevalence rates ranging from 80-420 cases per 100,000 population. The disease is commonly found throughout the Middle East, East Asian countries, and the Mediterranean basin, while becoming progressively rarer in Northern Europe and the Americas with a prevalence between 0.1-7.5 cases per 100,000. Beyond these defined conditions, many SAID patients present with undifferentiated autoinflammatory syndromes. A precise epidemiological estimate of these patients, however, presents a significant epidemiological challenge, with prevalence rates likely underreported globally due to inconsistent diagnostic criteria and recognition.”

- For sample selection, apart from the details correctly mentioned in the paper, did you perform an eventual power analysis or something similar, that could drive to a quantitative assessment of the impact of your investigation?

Yes, we have now added this as Supplementary Table 1, and added a comment about the power to the limitations of the study paragraph.

OR	Power	
	Without correction	Bonferroni correction
1.2	0.11	<0.01
1.5	0.38	0.01
1.8	0.66	0.07
2.0	0.80	0.15
2.5	0.96	0.44
3.0	0.99	0.72

- It is necessary to acknowledge the ethical approval(s) of the protocol followed by means of the number and name of the approving authority.

Agreed. We have now included the following in the methods:

“Consenting committees that approved the study were:

Ethics Committee of 14 French clinical sites: *Comité de Protection des Personnes Ouest 6* (EC Number: [19.01.03.64304], Approval Date: [08/04/2019]);

Ethics Committee: hacettepe university clinical research ethics boards (EC Number: [2020/08-09 (KA-20034)], Approval Date: [02/06/2020]);

Ethics Committee: *Commission Cantonale d’Ethique de la Recherche* (EC Number: [2020-01334], Approval Date: [02/11/2020]);

Ethics Committee: *CEIm Fundació Sant Joan de Déu* (EC Number: [PIC-112-19], Approval Date: [05/07/2019]);

Ethics Committee: Republic of Slovenia National Medical Ethics Committee (EC Number: [0120-495/2019/4], Approval Date: [15/11/2019]);

Ethics Committee: *Medische Ethische Toetsings Commissie* (EC Number: WT/ss/MEC-2020-0164], Approval Date: [112/10/2020]);

Ethics Committee: General hospital of Athens's board of directors (EC Number: [969/12-10-2020], Approval Date: [13/11/2020]);

Ethics Committee: Research UZ/KU Leuven (EC Number: [S63731], Approval Date: [12/08/2020]);

Ethics Committee: OPBG's Ethics Committee (EC Number: [1357], Approval Date: [24/11/2020]);

Ethics Committee: *Comitato Etico Milano Area 1* (EC Number: [2022/EM/060], Approval Date: [16/03/2022]);

Ethics Committee: *Comitato Etico Regionale per la Sperimentazione Clinica della Regione Toscana Sezione: AREA VASTA SUD EST* (EC Number: [19577], Approval Date: [21/02/2022]);

Ethics Committee: Comitato Etico per la Sperimentazione Clinica della Provincia di Padova (EC Number: [5350/AO/22], Approval Date: [10/03/2022]);

Ethics Committee: Medical Association of Westphalia-Lippe and the Westfälische Wilhelms-University of Münster (EC Number: [2020-476-f-S], Approval Date: [15/12/2020]);

Ethics Committee: South West - Frenchay Research Ethics Committee (EC Number: [

20/SW/0022276218], Approval Date: [18/08/2020])”

- As for the missing data management, how did you perform the imputation of data?

This is now described in the text, with the source code for the exact methodology available at <https://github.com/rafael-veiga/ImmunoAID.git>

“To address missing data, data was imputed, resulting in a reduction in data precision rather than data loss. K-nearest neighbors (KNN) imputation³⁵ was used to maintain data heterogeneity and provide good accuracy. Unlike imputation methods based on central tendency and regression models, KNN imputation does not significantly reduce data heterogeneity. Each cellular variant frequency exhibited its own unique distribution and magnitude. To prevent highly skewed distributions and large values from disproportionately influencing the distance measures, a Box-Cox transformation³⁶ was applied, followed by standardization prior to imputation. This process normalized the data, ensuring that the resulting dataset contained information more comparable in their metrics, for a more robust and consistent database for subsequent analyses.”

Reviewer #1

The authors have performed additional analyses in response to the previous review, which have substantially strengthened their dataset.

We thank the reviewer for acknowledging the value of the substantial additional experimental work included.

Major comments

1. Using cell-surface markers and newly performed proteomic profiling, the authors demonstrate similarities between autoinflammation of unknown origin (AUO) and Still's disease. However, could this AUO cohort in fact represent patients who should have been classified as having Still's disease? Although the Supplementary Table lists the AUO patients and their clinical features, the authors should specifically extract key Still's disease-related manifestations (e.g. high-grade periodic fever, salmon-pink rash, as defined in the ILAR criteria), together with distinctive features (e.g. urticaria, ulcers), present a focused comparison of these features, and clearly demonstrate that the AUO patients are indeed distinct from those with Still's disease.

All patients with Autoinflammation of unknown origin had Still's disease excluded by expert diagnostic review by the consulting clinician, both at inclusion and during a one-year follow-up period. We have now included in Supplementary Table 1 the Still's disease-inclusion criteria column, and additional columns for quotidian spiking fever and transient salmon-pink or evanescent erythematous rash, for the AUO patients. None of the AUO patients meet the Still's disease diagnosis criteria, nor did any have the salmon-pink rash or quotidian spiking fever. Urticaria was present in two patients and ulcers in one (data that was already included under "lesions" on the prior version). This information has been added to the methods section description of patient characteristics:

"All Autoinflammation of unknown origin patients failed to meet the diagnostic criteria of any alternative SAID, both at inclusion and during a one-year follow-up period. None of these patients presented with the characteristic Still's disease features of a quotidian spiking fever or transient salmon-pink or evanescent erythematous rash, and all demonstrated no mutations in the VEXAS syndrome-associated UBA1 gene."

2. The authors newly performed proteomic analyses; did classical Still's disease-related markers such as ferritin, IL-18, CD168, and CXCL9 differ between the AUO and Still's disease groups?

For comparison of selected markers, we performed additional quantification of ferritin, IL-18 and CXCL9. For each marker, AUO patients were significantly raised compared to healthy patients, and were intermediate to the levels observed in Still's disease. For ferritin and IL-18, the Still's disease patients were significantly elevated compared to AUO patients, for CXCL9, the Still's disease patients had a trend in elevation compared to AUO patients.

*"This trend was observed using diagnostic quantification of the classical Still's disease serum markers of ferritin, IL-18 and CXCL9, where patients with autoinflammation of unknown origin showed an intermediate phenotype between that of healthy and Still's disease patients (**Supplementary Figure 11**)."*

Supplementary Figure 11. Quantification of Still's disease biomarkers.

Quantification in plasma or serum of ferritin (Chemiluminescent Microparticle ImmunoAssay-based assay; ng/mL), IL-18 (Luminex; pg/mL), and CXCL9 (Luminex; pg/mL), in healthy controls and in patients diagnosed with autoinflammation of unknown origin (AUO), Still's disease, familial Mediterranean fever (FMF) or Behçet's disease. Statistical analyses were performed after assessing the normality of the data distribution. Depending on the outcome, either a one way ANOVA followed by Tukey's multiple comparisons test or a Kruskal–Wallis test followed by Dunn's multiple comparisons test was applied. Significance levels are indicated as follows: P < 0.05 (*), P < 0.01 (**), P < 0.001 (***), P < 0.0001 (****).

Minor comments

3. In Figures 1F and 2C, the arrows overlap with the text, making these panels difficult to read. The layout should be adjusted to avoid this overlap.

We have removed the box around the text and adjusted the location of the text to avoid overlap.

4. In Figure 2A, NK- and B-cell levels appear lower in the untreated group than in the treated group. The relevant significance levels (e.g. p-values) should be indicated. Given the large number of figures, the overall presentation is hard to follow; the authors should consider reducing the number of main figures and moving some panels to the Supplementary Material.

For Figure 2A and Figure 6A, there were no differences between treated vs untreated that reached the $p < 0.05$ threshold. We have now added that to the figure legend, rather than crowd the figure with non-significant values. We have considered carefully the data that is presented in the main figures vs the supplementary figures, and we believe the items in the main figures include the key messages of the paper, and prefer not to bury these in the already abundant supplementary material.

5. For Figures 1, 2, 4, 5, and 6, the corrected statistical significance levels, adjusted for the number of tests (multiple comparisons), should be indicated directly on the plots.

We have now done so, for the relevant comparisons: i.e., for disease vs healthy in Figure 1/4/5 and treated vs untreated in Figure 2/6. We have limited adding p values

to comparisons which reached the $p < 0.05$ threshold, rather than crowd the figure with non-significant values. A note on which comparisons are labelled with p values is given in each figure legend.

6. For Figure 3, the current presentation of all plots and ROC curves is difficult to interpret. It would be clearer if the authors could summarise these data in a table.

We have now added this as Table 1:

Table 1. Summary statistics for disease differences

Flow cytometry: average area under the receiver operating characteristic curve					
	Healthy	AUO	Behcet	FMF	Still's
Healthy	NA	0.92 (0.86-0.98)	0.77 (0.64-0.89)	0.78 (0.68-0.89)	0.94 (0.87-1.01)
AUO	0.92 (0.86-0.98)	NA	0.79 (0.64-0.93)	0.82 (0.67-0.96)	0.79 (0.67-0.91)
Behcet	0.77 (0.64-0.89)	0.79 (0.64-0.93)	NA	0.78 (0.64-0.93)	0.93 (0.85-1.01)
FMF	0.78 (0.68-0.89)	0.82 (0.67-0.96)	0.78 (0.64-0.93)	NA	0.93 (0.86-0.99)
Still's	0.94 (0.87-1.01)	0.79 (0.67-0.91)	0.93 (0.85-1.01)	0.93 (0.86-0.99)	NA

Flow cytometry: Euclidean distance between population averages					
	Healthy	AUO	Behcet	FMF	Still's
Healthy	NA	6.17	4.86	4.45	8.90
AUO	6.17	NA	6.59	6.12	5.14
Behcet	4.86	6.59	NA	4.26	8.87
FMF	4.45	6.12	4.26	NA	8.10
Still's	8.90	5.14	8.87	8.10	NA

Proteomics: average area under the receiver operating characteristic curve					
	Healthy	AUO	Behcet	FMF	Still's
Healthy	NA	0.94 (0.89-1.00)	0.85 (0.75-0.96)	0.93 (0.86-1.01)	0.97 (0.94-1.00)
AUO	0.94 (0.89-1.00)	NA	0.64 (0.45-0.83)	0.72 (0.59-0.86)	0.65 (0.46-0.84)
Behcet	0.85 (0.75-0.96)	0.64 (0.45-0.83)	NA	0.71 (0.54-0.88)	0.95 (0.89-1.00)
FMF	0.93 (0.86-1.01)	0.72 (0.59-0.86)	0.71 (0.54-0.88)	NA	0.90 (0.81-0.98)
Still's	0.97 (0.94-1.00)	0.65 (0.46-0.84)	0.95 (0.89-1.00)	0.90 (0.81-0.98)	NA

Proteomics: Euclidean distance between population averages					
	Healthy	AUO	Behcet	FMF	Still's
Healthy	NA	8.31	8.11	7.24	10.41
AUO	8.31	NA	7.24	5.98	5.83
Behcet	8.11	7.24	NA	5.93	9.95
FMF	7.24	5.98	5.93	NA	8.46
Still's	10.41	5.83	9.95	8.46	NA

7. In the Limitations section, the authors should also acknowledge that they did not assess RNA expression, and therefore the intracellular mechanisms underlying these findings remain insufficiently understood.

We have added the following point:

“Finally, flow cytometry and serum proteomics do not measure RNA expression, and thus intracellular changes reporting on key mechanistic pathways will have been missed.”

8. Some of the elderly male AUO patients may in fact have VEXAS syndrome. Have the authors performed somatic UBA1 genotyping in this cohort? If not, this should be mentioned in the discussion.

We have now performed somatic UBA1 genotyping in the AUO patients, and none carried a mutation. This is now included in the methods text.

“All Autoinflammation of unknown origin patients failed to meet the diagnostic criteria of any alternative SAID, both at inclusion and during a one-year follow-up period. None of these patients presented with the characteristic Still’s disease features of a quotidian spiking fever or transient salmon-pink or evanescent erythematous rash, and all demonstrated no mutations in the VEXAS syndrome-associated *UBA1* gene.”

Reviewer #2

I warmly thank the authors for having carefully and positively considered my suggestions. In my opinion, the manuscript is now more complete and its quality is higher than in the previous round. I just have a very minor remark, considering the thorough paragraph added about epidemiological features. In this regard, I would appreciate the inclusion of some bibliography to support the numbers included.

The thank the reviewer for their thoughtful and productive reviews throughout the revision process, and for their positive evaluation of our revised manuscript. We have now included citations to support the epidemiology paragraph, below:

“The prevalence of SAIDs ranges from 1 in 1000 to 1 in 1,000,000 people, depending on the specific disease, country, and population. Still's disease (encompassing both Adult-onset Still's Disease and systemic Juvenile Idiopathic Arthritis) presents a relatively uniform global distribution with an estimated prevalence of 1-10 cases and 10-100 cases per 1,000,000 population, for the adult and juvenile-onset form respectively, and occurring at similar rates across diverse populations worldwide^{6, 7, 8}. Familial Mediterranean Fever exhibits a pronounced ethnic and geographic concentration, with a significantly higher prevalence among specific Mediterranean and Middle Eastern populations. Prevalences vary between 1/250 (Sephardic Jews) and 1/2600 (Arabs). Significant prevalence rates are seen in countries like Greece, Italy, Spain, and Portugal, as well as in diaspora communities where affected populations have migrated^{9, 10}. Behçet's disease demonstrates a geographic pattern following the ancient Silk Road trading route, with Turkey showing the highest global prevalence rates ranging from 80-420 cases per 100,000 population. The disease is commonly found throughout the Middle East, East Asian countries, and the Mediterranean basin, while becoming progressively rarer in Northern Europe and the Americas with a prevalence between 0.1-7.5 cases per 100,000^{11, 12}. Beyond these defined conditions, many SAID patients present with undifferentiated autoinflammatory syndromes. A precise epidemiological estimate of these patients, however, presents a significant epidemiological challenge, with prevalence rates likely underreported globally due to inconsistent diagnostic criteria and recognition^{13, 14, 15}.”

6. Feist, E., Mitrovic, S. & Fautrel, B. Mechanisms, biomarkers and targets for adult-onset Still's disease. *Nat Rev Rheumatol* **14**, 603-618 (2018).
7. Giacomelli, R., Ruscitti, P. & Shoenfeld, Y. A comprehensive review on adult onset Still's disease. *J Autoimmun* **93**, 24-36 (2018).
8. Prakken, B., Albani, S. & Martini, A. Juvenile idiopathic arthritis. *Lancet* **377**, 2138-2149 (2011).

9. Ben-Chetrit, E. & Touitou, I. Familial mediterranean Fever in the world. *Arthritis Rheum* **61**, 1447-1453 (2009).
10. Touitou, I. The spectrum of Familial Mediterranean Fever (FMF) mutations. *Eur J Hum Genet* **9**, 473-483 (2001).
11. Alghamdi, M. & Lindsey, S. Behcet's disease unraveled: Insights into clinical manifestations, diagnosis, and management. *Medicine (Baltimore)* **104**, e44614 (2025).
12. Maldini, C., Druce, K., Basu, N., LaValley, M.P. & Mahr, A. Exploring the variability in Behcet's disease prevalence: a meta-analytical approach. *Rheumatology (Oxford)* **57**, 185-195 (2018).
13. Schnappauf, O. & Aksentijevich, I. Current and future advances in genetic testing in systemic autoinflammatory diseases. *Rheumatology (Oxford)* **58**, vi44-vi55 (2019).
14. Vanderschueren, S. *et al.* Inflammation of unknown origin versus fever of unknown origin: two of a kind. *Eur J Intern Med* **20**, 415-418 (2009).
15. Wright, W.F. *et al.* Recommendations for Updating Fever and Inflammation of Unknown Origin From a Modified Delphi Consensus Panel. *Open Forum Infect Dis* **11**, ofae298 (2024).

Reviewer #1

No additional comments.

Reviewer #2

All my concerns were successfully answered. Thank you.

We thank both reviewers for their constructive approach to our article